# Speeding Up Image Classifiers with Little Companions

## Abstract

Scaling up neural networks has been a key recipe to the success of large language and vision models. However, in practice, up-scaled models can be disproportionately costly in terms of computations, providing only marginal improvements in performance; for example, EfficientViT-L3-384 achieves <2% improvement on ImageNet-1K accuracy over the base L1-224 model, while requiring $14\times$ more multiply–accumulate operations (MACs). In this paper, we investigate scaling properties of popular families of neural networks for image classification, and find that scaled-up models mostly help with "difficult" samples. Decomposing the samples by difficulty, we develop an embarrassingly simple model-agnostic **two-pass** Little-Big algorithm that first uses a light-weight "little" model to make predictions of all samples, and only passes the difficult ones for the "big" model to solve. Good little companions achieve drastic MACs reduction for a wide variety of model families and scales. Without loss of accuracy or modification of existing models, our Little-Big models achieve MACs reductions of 76% for EfficientViT-L3-384, 81% for EfficientNet-B7-600, 71% for DeiT3-L-384 on ImageNet-1K. Little-Big also speeds up the InternImage-G-512 model by 62% while achieving 90% ImageNet-1K top-1 accuracy, serving both as a strong baseline and as a simple practical method for large model compression.

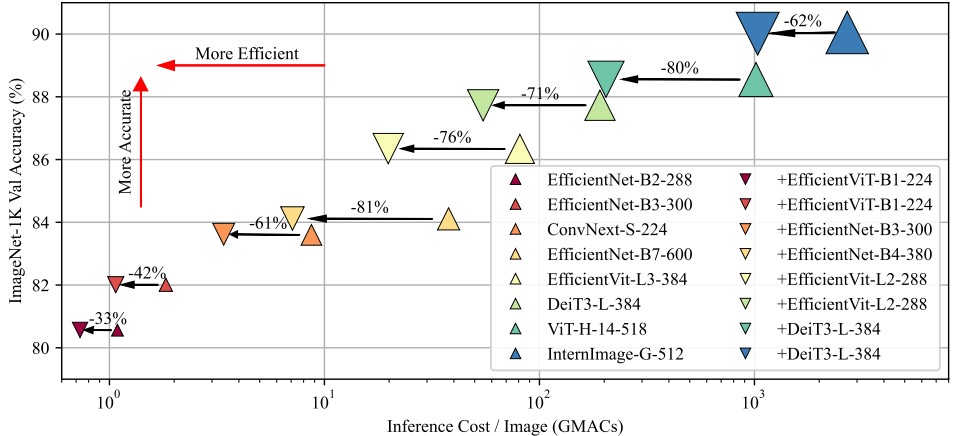

**Figure 1:** Little-Big relaxes the assumption of obtaining predictions for samples in a single pass using a single model, achieving MACs reduction of $30\% - 80\%$ across models types (convolutional neural networks, transformers, and hybrid networks) and scales (from 1 to 3000 GMACs). Marker sizes correspond to $log(\#parameters)$. Model labels are formatted as "Family-Size-InputResolution".

# 1 INTRODUCTION

Advances in parallel computing hardware, such as GPUs, have made end-to-end single-pass parallel processing standard in computer vision models. Large vision datasets like ImageNet-1K (Deng et al., 2009) made it possible for such deep vision models (e.g.Alexnet (Krizhevsky et al., 2012), ResNet (He et al., 2016) and ViT Dosovitskiy et al. (2020)) to learn general visual features at scale. While vision models surpassed human performance on ImageNet-1K a decade ago (Szegedy et al., 2015), researchers are in the perpetual pursuit of achieving improved performance by using a combination of two strategies: 1) developing more performant models and training techniques for a given compute budget, and 2) scaling up the models. While improved models from the former approach are often preferred in compute or memory-constrained applications, the latter has become increasingly popular, thanks to its success in large language models (LLMs) (Touvron et al., 2023). However, despite architectural improvements, scaling up models remains expensive; we are often trading exponential compute cost for marginal gains in model accuracy (Table 1).

In this work, we show that much of the inefficiency comes from our implicit preference for **single-pass** models and propose an embarrassingly simple **two-pass** algorithm to drastically speed up models with little companions. We summarize our work in response to two critical questions around model scaling and compression.

⟨**Q1**⟩ *Given a pair of Little-Big models in the same model family, which incorrect predictions made by the Little model are fixed using the Big model?*

- Using ImageNet-1K as the test bed and binning the Little model's predictions by confidence, we find that, very often, mistakes made by the Little model correspond to low confidence (measured via maximum softmax probability)

⟨**Q2**⟩ *Without compromising accuracy, can we speed up a Big model by using a Little model to preprocess a proportion of samples in a distribution?*

- We propose an embarrassingly simple two-pass Little-Big protocol where a light-weight **Little** model is used to make predictions (class and confidence) on all samples in the first pass, and a **Big** model performs a second pass on samples with low confidence from the first pass, achieving significant reduction in inference compute costs without compromising accuracy.

- Without any modification to existing models, we prescribe Little-Big pairs that significantly reduce compute costs of models across types and scales, while not compromising accuracy: Little-Big models achieves MACs reductions of 76% for EfficientViT-L3-384, 81% for EfficientNet-B7-600, 71% for DeiT3-L-384 on ImageNet-1K. Little-Big also speeds up the very large InternImage-G-512 model by 62% while achieving 90% ImageNet-1K top-1 accuracy, serving both as a strong baseline and as a practical approach for efficient deployment of large models.

**Table 1: Scaling up is expensive.** Scaling up model size is a popular way to improve performance without redesigning neural architectures or training recipes. Popular practices often involve sparing compound scaling in input resolution $H$, model width $w$ and depth $l$ over the base model (characterized by $H_0, w_0, l_0$). However Equation 2 shows that model size and inference cost quickly blow up by $10 - 100\times$, with only marginal performance gains over small base models of the same family.

| Model Family | Size | $H/H_0$ | $w/w_0$ | $l/l_0$ | ImageNet-1K Val | | GMACs | |
|---|---|---|---|---|---|---|---|---|
| | | | | | Accuracy (%) | $\Delta$ | Absolute | $\Delta$ |
| EfficientNet (Tan & Le, 2019) | B0-224 | 1 | 1 | 1 | 77.65 | $--$ | 0.39 | $--$ |
| | B2-280 | 1.3 | 1.1 | 1.2 | 80.56 | +2.91 | 1.09 | +1.8× |
| | B4-380 | 1.7 | 1.4 | 1.8 | 83.45 | +5.80 | 4.39 | +10.3× |
| | B7-600 | 2.7 | 2.0 | 3.1 | 84.11 | +6.45 | 37.8 | +95.8× |
| EfficientViT (Cai et al., 2023) | L1-224 | 1 | 1 | 1 | 84.39 | $--$ | 5.3 | $--$ |
| | L2-288 | 1.3 | 1 | 1.4 | 85.60 | +1.21 | 11.0 | +1.1× |
| | L3-384 | 1.7 | 2 | 1.4 | 86.34 | +1.95 | 81.0 | +14.3× |
| DeiT3 (Touvron et al., 2022) | S-224 | 1 | 1 | 1 | 83.05 | $--$ | 4.6 | $--$ |
| | B-224 | 1 | 2 | 1 | 85.60 | +2.55 | 17.6 | +1.1× |
| | L-384 | 1.7 | 2.7 | 2 | 87.73 | +4.68 | 191.2 | +40.6× |

## 2 RELATED WORK

### 2.1 COMPUTER VISION MODELS

Since Alexnet (Krizhevsky et al., 2012), single-pass neural classifiers trained end-to-end have dominated leaderboards of various vision tasks from image classification to video segmentation (Beyer et al., 2020; Deng et al., 2009; He et al., 2016; Szegedy et al., 2015; Xu et al., 2018). Most neural models originates from two families of neural architectures: convolutional neural networks (CNNs) and transformers. Core to CNNs are "convolutions" which apply the same compute across locations on a feature map. On the other hand, transformers, which first found success in sequence-to-sequence language models (Vaswani et al., 2017) and subsequently in vision (Dosovitskiy et al., 2020), embed a sequence of tokens (e.g., image patches) and utilize attention mechanism to model intra- and inter-token interactions. Finally, hybrid models like Swin (Liu et al., 2021) combine CNN priors and attention mechanisms to achieve good performance.

### 2.2 SCALING IN VISION

Many neural classifiers can be expressed as a composition of layers (He et al., 2016; Krizhevsky et al., 2012; Szegedy et al., 2015; Touvron et al., 2022):

$$y = F(x) = f_{w_l} \cdot ... \cdot f_{w_1} \cdot f_{w_0}(x), \tag{1}$$

where $x \in \mathbb{R}^{C \times H \times H}$ denotes an input image (using square images for simplicity) and $y \in (0,1)^N$ denotes a $N$-dimensional softmax confidence score. To get a class prediction, one finds the class $n$ with the highest confidence. $f_{w_j}$ denotes the function of layer $j$ with its characteristic width $w_j$. The inference cost of a single sample $x$ with such a model $F(x)$ can be expressed as:

$$MACs[F(x)] \approx C_F * H^2 * w^2 * l \tag{2}$$

where $C_F$ is a scaling coefficient determined by the model family Tan & Le (2019), and $w$ denotes the model width, while $l$ represents its depth. In turn, the average inference cost per sample over a (finite) distribution $D$ is given by:

$$\mathbb{E}_D\left(MACs[F]\right) = \frac{1}{|D|} \sum_{x \in D} MACs[F(x)] \approx C_F * H^2 * w^2 * l \tag{3}$$

Complementing innovations in model architecture $F$ that make models more compute efficient, a straightforward way of improving performance is to scale up the model. Thanks to architectural improvements like the skip connections (He et al., 2016), normalization layers Szegedy et al. (2015), as well as better initialization and parameterization (Yang & Hu, 2021), scaling up a model by orders of magnitude has been made feasible. Furthermore, efficient scaling strategies such as compound scaling in model width and depth, as well as input resolution (Tan & Le, 2019) have also emerged. However, Equation 2 imposes a fundamental limitation on the prohibitive cost of scaling: doubling in $H$, $w$, and $l$ leads to $2^5 - 1 = 31\times$ increase in model compute cost. The marginal accuracy gains associated with model scaling shown in Table 1 further make it unappealing for many practical use cases with limited compute budget.

### 2.3 MODEL COMPRESSION

It is well known that modern neural networks have significant redundancy, thus impacting both the **model size** (often measured by the number of parameters) and inference **compute cost** (often measured by MACs). This has motivated the design of compression strategies to reduce redundancy in the model. This has motivated the development of various compression strategies to reduce redundancy. Popular approaches include:

- **Pruning**: Ablating weights or neurons deemed less important for predictions. For example, WDPruning (Yu et al., 2022a) reduces width based on saliency scores, while X-Pruner (Yu & Xiang, 2023) measures a unit's importance by its contribution to predicting each target class. The underlying ideas of these models is that the redundancy in size and compute are coupled, and one reduces compute by removing the units.

- **Adaptive Computation**: Employing mechanisms that dynamically adjust the computation based on input complexity (Kag & Fedorov, 2023; Rao et al., 2021; Yin et al., 2022). These methods often adopt some form of "early exit" mechanisms that reduce $l$ in Equation 2. For instance, techniques like DynamicViT (Rao et al., 2021) downsamples the number of tokens adaptively to reduce compute cost, while A-ViT (Yin et al., 2022) introduces learned token halting for ViT models so that not all tokens are processed by the full depth of the model, thus effectively reducing the average depth of the compute graph during inference.

- **Quantization**: This is a widely adopted technique that reduces the precision of model weights and activations, thereby decreasing memory usage and computational requirements (Rokh et al., 2023). While quantization is not the focus of Little-Big, it is worth noting that it can complement our framework to further reduce inference costs. For instance, quantized versions of both Little and Big models could be used within the Little-Big framework, offering compounded efficiency gains. However, it is also important to highlight that the benefits of quantization are often hardware-dependent, relying on specific accelerators or processors that support low-precision computations. In contrast, Little-Big is entirely hardware-independent, making it applicable across a wide range of deployment environments without requiring specialized hardware.

## 2.4 HUMAN VISION

While parallel processing plays an essential role in making it possible to ingest gigabits/s of raw visual information and compress it to tens of bits/s to guide our behavior (Soto et al., 2020; Zheng & Meister, 2024), human vision is **not a single-pass process**. Human eyes have two distinct information processing pathways that originate from two types of photoreceptors called rods and cones. While the visual acuity in the cone-rich fovea is $\sim 1$ arcmin, it only covers $\sim 2$ degrees, or $\sim 0.01\%$ of the visual field (Rosenholtz, 2016). The rest of the $\sim 99.9\%$ of the visual field is mostly dominated by rods which provides much lower visual acuity ($\sim 10$ arcmin). A given small patch in the visual field either only gets processed in a single-pass by the low-acuity rod pathway, or followed by additional passes with high-acuity foveal vision if needed as directed by the saccade. Studies (Tang et al., 2018; Torralba, 2009) have shown that human visual classification performance adapts to variable compute budget.

## 3 SCALING UP HELPS WITH "HARD" SAMPLES

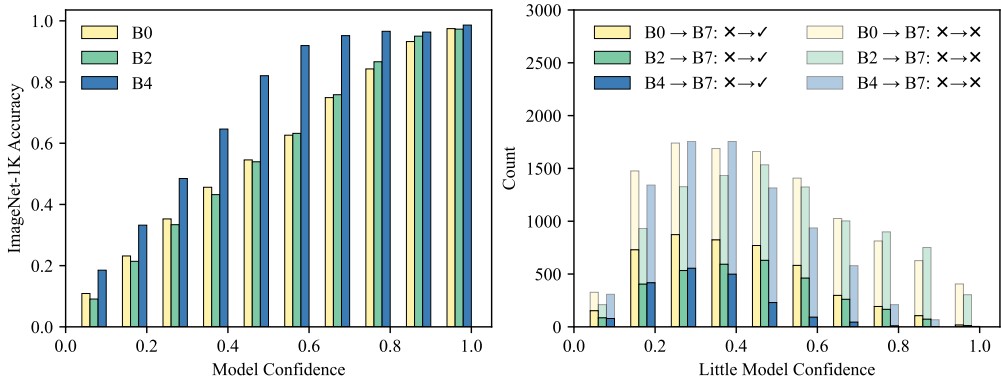

**Figure 2:** Using the EfficientNet family as an example, we first show that confidence of individual models correlate well with prediction accuracy(**left**). This allows us to approximate a **"hardness"** axis, where harder samples correspond to predictions with lower confidence. Mistakes made by Little models can be categorized as *correctable* (solid bars, $\times \rightarrow \checkmark$) and *non-correctable* (shaded bars, $\times \rightarrow \times$) by the Big model. Breaking down the mistakes of Little models by hardness in the **right** panel, we find that most of the correctable mistakes are characterized by low confidence. This motivates the idea of a two-pass Little-Big algorithm enabled by decomposing the samples by confidence thresholds. More examples can be found in Appendix Figure 6.

In answer to $\langle$**Q1**$\rangle$, we first define an axis of "hardness" along which we can break down the predictions of the Little models. In lieu of an objective notion of hardness, we use the model confidence $max(F_{small}(x_i))$ as a surrogate since it reflects model calibration, i.e., higher prediction confidence correspond to higher accuracy (Figure 2 left).

Intuitively, there are 3 simple hypotheses on how scaled up "Big" models help correct the mistakes of base "Little" models:

> $\langle$**H1**$\rangle$ *Big models uniformly help samples across difficulty,*

> $\langle$**H2**$\rangle$ *Big models preferentially help with "hard" samples,*

> $\langle$**H3**$\rangle$ *Big models preferentially help with "easy" samples.*

Using the Pytorch (Paszke et al., 2019) pretrained EffcientNet family (Tan & Le, 2019) as an example, the right panel in Figure 2 visualizes prediction mistakes by Little model confidence (B0, B2, and B4) on the ImageNet-1K validation set. The mistakes made by Little models are divided into two categories, *correctable* (solid bars) and *non-correctable* (shaded bars) by the Big model. The full height of each bar (solid + shaded parts) sums up to the total number mistakes in the corresponding bin. Quantitatively, the average confidence of *correctable* mistakes for EfficientNet-B0+B7, EfficientNet-B2+B7, EfficientNet-B4+B7 pairs are $0.38$, $0.41$, and $0.30$ respectively. In fact, $90\%$ of correctable mistakes fall under confidence thresholds of $0.65$, $0.67$, $0.47$, respectively. This suggests $\langle$**H2**$\rangle$ is the likely to be true and motivates the two-pass algorithm in the next section.

## 4 TWO-PASS LITTLE-BIG ALGORITHM

### 4.1 SPEEDING UP BIG MODELS

```python
import torch.nn.functional as F

Class BigLittle:

    __init__(self,
             little,    # small model
             big,       # big model
             t_little,  # image transform for small model
             t_big,     # image transform for big model
             ):

        self.little, self.big   = little, big
        self.t_little, self.t_big = t_little, t_big

    predict(self,
            x,          # raw input image
            threshold   # prediction threshold T
            ):

        y = F.softmax(self.little(self.t_little(x)),dim=1)

        if torch.max(y) < threshold:
            y = F.softmax(self.big(self.t_big(x)),dim=1)

        return F.argmax(y,dim=1)
```

**Little-Big Algorithm**: Pytorch pseudo code of Little-Big for single image inference. Separate pre-processing image transforms are included as Big and Little models may require different input resolution and/or interpolation. The implementation keeps both models in the memory, Big and Little models can be loaded/unloaded to avoid overhead in max memory usage.

To answer $\langle$**Q2**$\rangle$, when we allow a sample to be solved with more than one pass like human vision, a simple way is to have a Little-Big pair $G_{F_{Little},F_{Big}}(x,T)$ or shortly $G(x,T)$ for simplicity:

$$G_{F_{Little},F_{Big}}(x,T) = \begin{cases} F_{Little}(x), \text{if } max(F_{Little}(x)) \geq T \\ F_{Big}(x). \end{cases} \qquad \textbf{Little-Big} \qquad (4)$$

The essence of this Little-Big algorithm is to use a light-weight model to pre-screen samples and only pass hard samples to the Big model (see Algorithm). The average per-sample cost of inference over a dataset $D$ can then be expressed as:

$$\mathbb{E}_D(MACs[G(x,T)]) = \frac{1}{|D|}\Bigg[\sum_{x \sim D} MACs[F_{Little}(x)] + \sum_{x \sim D^*} MACs[F_{Big}(x)]\Bigg]$$

$$= MACs[F_{Little}(x)] + \frac{|D^*|}{|D|}MACs[F_{Big}(x)], \tag{5}$$

where $D^* \subseteq D$ is defined as the set of $x$ where $max(F_{Little}(x)) < T, \forall x \in D$.

Using EfficientNet-B7 (Paszke et al., 2019) as an example Big model to speed up, the top left panel in Figure 3 shows how the relative size $|D^*|/|D|$ varies as a function of threshold $T$ with Little models ranging from EfficientNet-B0 to B6. The shape of the curves correspond to the cumulative distribution of prediction confidence for each Little model. Equation 5 further links $|D^*|/|D|$ to the relative compute cost $\mathbb{E}_D(MACs[G(x,T)])/\mathbb{E}_D(MACs[F_{Big}(x)])$: the higher the threshold the more samples that get passed to the big model, thus increasing MACs. Since $|D^*|/|D| \leq 1$, the upper bound of relative MACs overhead in the worst case scenario is $MACs[F_{Little}(x)]/MACs[F_{Big}(x)]$, which is usually no greater than 1 with proper choices of the Little model. As shown in the top middle panel of Figure 3, the net effect of adding a pre-screening Little model to the Big model leads to significant reduction in compute cost for a wide range of $T$ across difference choices of Little models.

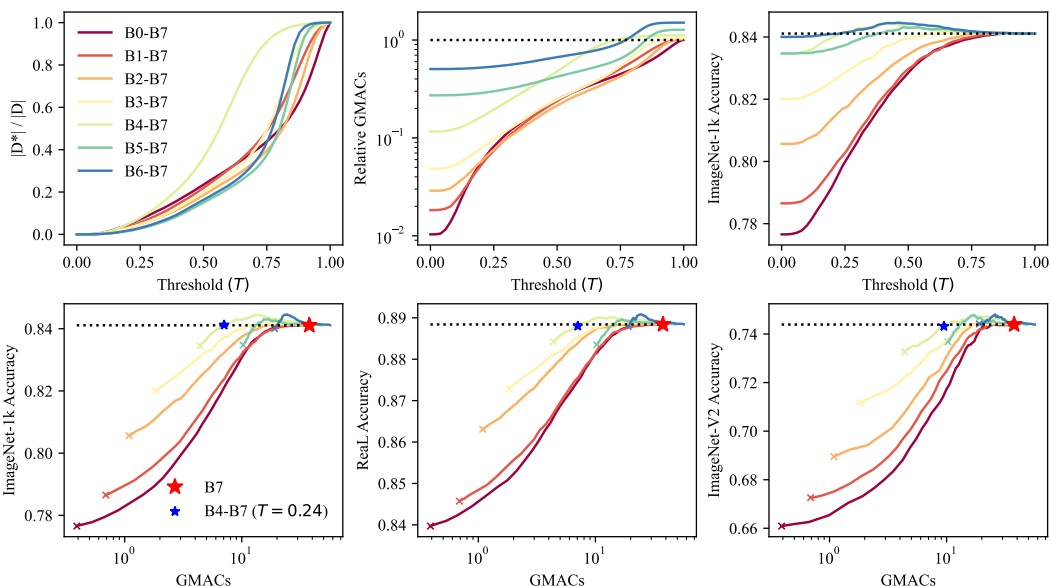

**Figure 3: Speeding up EfficientNet-B7 with smaller EfficientNets.** For using Little-Big, one needs to choose what small model to use and set a threshold $T$ based on an accuracy or MACs target. 50 evenly spaced $T$ in the range of 0 to 1 are sampled to generate each curve. "★" marks the accuracy-MACs tradeoff of the Big EfficientNet-B7. "★" indicates the optimal Little-Big pair without any loss of accuracy on ImageNet-1K, achieving 81% of MACs reduction. The same fixed $T$ performs well on both ReaL and ImageNet-V2 as well. Importantly, the optimal pair achieves both better accuracy and lower MACs than simply scaling down B7 to B6 or B5.

## 4.2 Speeding Up Without Losing Accuracy

**Accuracy-MACs trade-off** In addition to MACs, one is interested in how the accuracy of $G(x,T)$ changes as a function of $T$. While it is not surprising that the accuracy generally decreases with smaller $T$, eventually degenerating to the accuracy of the Little model at $T = 0$, certain choices of $G(x,T)$ achieve very little accuracy drop or even slight accuracy boost across a wide range of $T$. The bottom left panel of Figure 3 visualizes the accuracy-MACs trade-off on ImageNet-1K. One might immediately notice that **any point** on the curves is a valid $G(x,T)$ model, and adjusting $T$ allows traversal along the full curve. Many points on certain curves, such as $G_{B4,B7}(x,T)$, achieves

new Pareto optima. A convenient way to pick an optimal operating point of $G(x, T)$ is to set up a target accuracy either in the form of absolute accuracy or tolerable accuracy loss $\Delta Acc$. Setting $\Delta Acc >= 0$, one can easily draw a horizontal line on the accuracy-MACs plot (dashed line in bottom row of Figure 3) and find the leftmost intersection point (blue star, $G_{B4,B7}(x, T = 0.24)$), yielding the Little-Big pair with the least compute cost while satisfying the accuracy target, speeding up B7 by $81\%$. Little-Big also achieves significant speed-up in throughput and latency (Appendix Figure 9).

**Generalization beyond ImageNet-1K** To test whether the optimal $G$ determined on ImageNet-1K generalizes well, individual models as well as Little-Big pairs are evaluated two additional datasets: 1) ImageNet-ReaL (Beyer et al., 2020) where the ground truth labels of the ImageNet-1K validation set are reassessed with an improved labeling protocol ((Figure 3 bottom middle), and 2) ImageNet-V2 (Recht et al., 2019) where slightly "harder" validation images are collected (Figure 3 bottom right). Comparing accuracy-MACs curves across the three datasets, one may notice that although the absolute accuracies and MACs varies, the shapes match qualitatively. Quantitatively, we find the optimal $G_{B4,B7}(x, T = 0.24)$ found on ImageNet-1K performs well on ReaL and V2, with only marginal accuracy losses of $0.04\%$ and $0.07\%$, respectively, validating the generalizability of the optimal $G$.

Evidently, the key to determining the optimal $G$ is estimating the accuracy-MACs trade-off curves on the target distribution $D$. In practice, however, one might only have access to a small subset of $D$ or a set $D'$ close to to $D$. We simulate this case by determining an optimal $G(x, T)$ on the smaller V2 (10000 samples) and test generalization performance on ImageNet-1K and ReaL (50000 samples). Following the aforementioned process of determining optimal $G(x, T)$ on V2 (Appendix Figure 7) yields a similar optimal pair $G'_{B4,B7}(x, T = 0.28)$), speeding up B7 by a similar $78\%$ on ImageNet-1K. This validates the robustness of such a process.

### 4.3 Speeding Up Models Across Types And Scales

Table 2 shows more examples the strong MACs compression with Little-Big on a variety of model families including CNNs, transformers, and hybrid models, across scales from 1 to 2700 GMACs. Thanks to the model agnostic nature of Little-Big , it allows pairing up models of different families (models in blue in Table 2). Large modes such as EfficientVit-L3-384 (Cai et al., 2023), DeiT3-L3-384 (Touvron et al., 2022), ViT-H-14-518 (Dosovitskiy et al., 2020) can be compressed without any loss of accuracy by $70\%$ to $80\%$.

It is worth noting that the roles of Big and Little in the Little-Big pair are **relative**. For example, DeiT3-L-384 performs well as Little model, efficiently speeding up the 3B-parameter InternImage-G-512 (Wang et al., 2022b) and 600M-parameter ViT-H-14-518 by $62\%$ and $80\%$, respectively. However, it does not imply that DeiT3-L-384 itself cannot act as the Big model and be sped up by an even smaller model. In fact, Table 2 shows that it can be sped up by the EfficientViT-L2-288 by $71\%$. Similarly, EfficientNet-B4 can serve as a strong Little model for compressing EfficientNet-B7, as well as a Big model to be compressed by the smaller EfficientNet-B2. This enables the generalization of Little-Big to a $K$-pass framework where $K > 1$ with additional performance gain (Appendix Section A.2). Another important observation is that the speed up depends on the distribution $D$: Relative MACs reduction are consistently lower on the slightly harder ImageNet-V2 than on ImageNet-1K. This suggests that one may measure distribution shift by measuring average MACs with Little-Big. Furthermore in Appendix A.3, we investigate a more sophisticated variant of the Little-Big approach, termed Little-Big Ensemble, where predictions from the Little and Big models are combined using a weighted average for hard samples. Our findings reveal that despite the added memory overhead of storing the Little model's predictions, Little-Big Ensemble does not yield significant improvements in achieving lossless accuracy compared to the simpler Little-Big method.

### 4.4 Comparison with Prior Art

As discussed in Section 2.3, many methods have been developed for model compression in various axis. We compare accuracy-MACs tradeoff of Little-Big with many such methods in Table 3. Typical pruning methods such as WDPruning (Yu et al., 2022a) and X-Pruner (Yu & Xiang, 2023) selectively removes units that are less important to model accuracy, methods like SPViT (He et al., 2024) "prune" some attention layers into convolutional layers, effecitvely changing the model type. Addtionally,

**Table 2: The simple Little-Big algorithm achieves strong MACs reduction across model types and scales.** Thresholds ($T$) are determined as the minimum value achieving a preset accuracy loss tolerance on ImageNet-1K (IN-1K). All but InterImage-G-512 are compressed without loss in IN-1K accuracy. Models in blue denote the Little model is from a different model family as the Big model. Configurations with lowest MACs are in **bold**.

| Model | Params | Top@1 Accuracy (%) | | | GMACs | |
|---|---|---|---|---|---|---|
| | | IN-1k | ReaL | V2 | IN-1k&ReaL | IN-V2 |
| Efficientnet-B2-288 (Tan & Le, 2019) | $9.1M$ | $80.56$ | $86.31$ | $68.95$ | $1.09$ | |
| +B0-224 $_{(T=0.66)}$ | $+3.5M_{(+58\%)}$ | $80.59_{(+0.03)}$ | $86.35_{(+0.04)}$ | $68.99_{(+0.04)}$ | $0.78_{(-28\%)}$ | $0.91_{(-16\%)}$ |
| +EfficientViT-B1-224 $_{(T=0.58)}$ | $+9.1M_{(+100\%)}$ | $80.57_{(+0.01)}$ | $86.35_{(+0.04)}$ | $68.92_{(-0.03)}$ | $\mathbf{0.73_{(-33\%)}}$ | $\mathbf{0.83_{(-24\%)}}$ |
| Efficientnet-B3-300 (Tan & Le, 2019) | $12.2M$ | $82.01$ | $87.28$ | $71.16$ | $1.83$ | |
| +B1-240 $_{(T=0.66)}$ | $+7.8M_{(+64\%)}$ | $82.01_{(+0.00)}$ | $87.35_{(+0.07)}$ | $71.15_{(-0.01)}$ | $1.36_{(-26\%)}$ | $1.56_{(-15\%)}$ |
| +EfficientViT-B1-224 $_{(T=0.78)}$ | $+9.1M_{(+74\%)}$ | $82.01_{(+0.00)}$ | $87.31_{(+0.03)}$ | $70.94_{(-0.22)}$ | $\mathbf{1.07_{(-42\%)}}$ | $\mathbf{1.27_{(-31\%)}}$ |
| Efficientnet-B4-380 (Tan & Le, 2019) | $19.3M$ | $83.45$ | $88.43$ | $73.27$ | $4.6$ | |
| +B3-300 $_{(T=0.50)}$ | $+12.2M_{(+63\%)}$ | $83.46_{(+0.01)}$ | $88.42_{(-0.01)}$ | $73.39_{(+0.12)}$ | $2.7_{(-38\%)}$ | $\mathbf{3.1_{(-29\%)}}$ |
| +B2-288 $_{(T=0.72)}$ | $+9.1M_{(+47\%)}$ | $83.47_{(+0.02)}$ | $88.42_{(-0.01)}$ | $73.28_{(+0.01)}$ | $\mathbf{2.7_{(-39\%)}}$ | $3.2_{(-27\%)}$ |
| +B1-240 $_{(T=0.86)}$ | $+7.8M_{(+40\%)}$ | $83.46_{(+0.01)}$ | $88.44_{(+0.01)}$ | $73.27_{(+0.00)}$ | $4.0_{(-10\%)}$ | $4.3_{(-3\%)}$ |
| +B0-224 $_{(T=0.94)}$ | $+5.3M_{(+27\%)}$ | $83.45_{(+0.00)}$ | $88.43_{(+0.00)}$ | $73.29_{(+0.02)}$ | $3.9_{(-10\%)}$ | $4.2_{(-4\%)}$ |
| EfficientViT-B3-288 (Cai et al., 2023) | $48.6M$ | $84.13$ | $88.49$ | $74.12$ | $6.5$ | |
| +B3-224 $_{(T=0.60)}$ | $+48.6M_{(+100\%)}$ | $84.14_{(+0.01)}$ | $88.49_{(+0.00)}$ | $73.98_{(-0.14)}$ | $4.7_{(-28\%)}$ | $5.1_{(-22\%)}$ |
| +B2-288 $_{(T=0.76)}$ | $+24.3M_{(+50\%)}$ | $84.13_{(+0.00)}$ | $88.55_{(+0.06)}$ | $73.82_{(-0.30)}$ | $\mathbf{3.8_{(-42\%)}}$ | $\mathbf{4.3_{(-33\%)}}$ |
| +B2-224 $_{(T=0.94)}$ | $+24.3M_{(+50\%)}$ | $84.14_{(+0.01)}$ | $88.52_{(+0.03)}$ | $74.11_{(-0.01)}$ | $\mathbf{3.8_{(-42\%)}}$ | $4.5_{(-30\%)}$ |
| ConvNext-L-224 (Liu et al., 2022) | $197.8M$ | $84.39$ | $88.75$ | $74.34$ | $34.3$ | |
| +S-224 $_{(T=0.52)}$ | $+50.2M_{(+25\%)}$ | $84.39_{(+0.00)}$ | $88.75_{(+0.00)}$ | $74.35_{(+0.01)}$ | $15.7_{(-54\%)}$ | $19.2_{(-44\%)}$ |
| Efficientnet-B7-600 (Tan & Le, 2019) | $66.3M$ | $84.11$ | $88.84$ | $74.39$ | $37.8$ | |
| +B6-528 $_{(T=0.24)}$ | $+43.0M_{(+65\%)}$ | $84.13_{(+0.02)}$ | $88.90_{(+0.06)}$ | $74.50_{(+0.11)}$ | $20.1_{(-47\%)}$ | $21.4_{(-43\%)}$ |
| +B5-456 $_{(T=0.38)}$ | $+30.4M_{(+46\%)}$ | $84.12_{(+0.01)}$ | $88.78_{(-0.06)}$ | $74.65_{(+0.26)}$ | $13.2_{(-65\%)}$ | $15.4_{(-59\%)}$ |
| +B4-380 $_{(T=0.24)}$ | $+19.3M_{(+29\%)}$ | $84.12_{(+0.01)}$ | $88.80_{(-0.04)}$ | $74.32_{(-0.07)}$ | $\mathbf{7.1_{(-81\%)}}$ | $\mathbf{9.5_{(-75\%)}}$ |
| +B3-300 $_{(T=0.66)}$ | $+12.2M_{(+18\%)}$ | $84.13_{(+0.02)}$ | $88.88_{(+0.04)}$ | $74.41_{(+0.02)}$ | $14.6_{(-61\%)}$ | $18.9_{(-50\%)}$ |
| +B2-288 $_{(T=0.74)}$ | $+9.1M_{(+14\%)}$ | $84.11_{(+0.00)}$ | $88.83_{(-0.01)}$ | $74.39_{(+0.00)}$ | $15.6_{(-59\%)}$ | $20.1_{(-47\%)}$ |
| +B1-240 $_{(T=0.90)}$ | $+7.8M_{(+12\%)}$ | $84.12_{(+0.01)}$ | $88.85_{(+0.01)}$ | $74.34_{(-0.05)}$ | $32.9_{(-13\%)}$ | $34.6_{(-8\%)}$ |
| +B0-224 $_{(T=0.92)}$ | $+5.3M_{(+8\%)}$ | $84.12_{(+0.01)}$ | $88.84_{(+0.00)}$ | $74.40_{(+0.01)}$ | $28.4_{(-25\%)}$ | $31.1_{(-18\%)}$ |
| EfficientViT-L3-384 (Cai et al., 2023) | $246.0M$ | $86.34$ | $89.66$ | $77.35$ | $81.1$ | |
| +L3-256 $_{(T=0.52)}$ | $+246.0M_{(+100\%)}$ | $86.35_{(+0.01)}$ | $89.71_{(+0.05)}$ | $77.36_{(+0.01)}$ | $40.4_{(-50\%)}$ | $44.3_{(-45\%)}$ |
| +L2-384 $_{(T=0.60)}$ | $+63.7M_{(+26\%)}$ | $86.34_{(+0.00)}$ | $89.83_{(+0.17)}$ | $77.55_{(+0.20)}$ | $27.0_{(-67\%)}$ | $31.7_{(-61\%)}$ |
| +L2-288 $_{(T=0.66)}$ | $+63.7M_{(+26\%)}$ | $86.35_{(+0.01)}$ | $89.86_{(+0.20)}$ | $77.37_{(+0.02)}$ | $\mathbf{19.8_{(-76\%)}}$ | $\mathbf{25.0_{(-69\%)}}$ |
| DeiT3-L-384 (Touvron et al., 2022) | $304.8M$ | $87.73$ | $90.24$ | $79.34$ | $191.2$ | |
| +B-224 $_{(T=0.82)}$ | $+86.6M_{(+28\%)}$ | $87.73_{(+0.00)}$ | $90.23_{(-0.01)}$ | $79.36_{(+0.02)}$ | $71.2_{(-63\%)}$ | $90.3_{(-53\%)}$ |
| +EfficientViT-L2-288 $_{(T=0.90)}$ | $+63.7M_{(+21\%)}$ | $87.73_{(+0.00)}$ | $90.37_{(+0.13)}$ | $79.43_{(+0.09)}$ | $\mathbf{54.7_{(-71\%)}}$ | $\mathbf{74.0_{(-61\%)}}$ |
| ViT-H-14-518 (Dosovitskiy et al., 2020) | $633.5$ | $88.55$ | $90.51$ | $81.12$ | $1016$ | |
| +L-16-512 $_{(T=0.46)}$ | $+305M_{(+48\%)}$ | $88.59_{(+0.04)}$ | $90.89_{(+0.38)}$ | $81.04_{(-0.08)}$ | $430_{(-58\%)}$ | $488_{(-52\%)}$ |
| +DeiT3-L-384 $_{(T=0.70)}$ | $+305M_{(+10\%)}$ | $88.56_{(+0.01)}$ | $90.64_{(+0.13)}$ | $81.06_{(-0.06)}$ | $\mathbf{204_{(-80\%)}}$ | $\mathbf{276_{(-73\%)}}$ |
| InternImage-G-512 (Wang et al., 2022b) | $3.076B$ | $90.05$ | $90.97$ | $83.04$ | $2700$ | |
| +XL-384 $_{(T=0.84)}$ | $+335M_{(+11\%)}$ | $90.01_{(-0.04)}$ | $90.98_{(+0.01)}$ | $82.99_{(-0.05)}$ | $1436_{(-47\%)}$ | $1781_{(-34\%)}$ |
| +DeiT3-L-384 $_{(T=0.90)}$ | $+305M_{(+10\%)}$ | $90.03_{(-0.02)}$ | $90.99_{(+0.02)}$ | $82.95_{(-0.09)}$ | $\mathbf{1035_{(-62\%)}}$ | $\mathbf{1335_{(-51\%)}}$ |

distillation is used to retrain the network to achieve better performance. We show that even with tricks that effectively retrained models, many pruning methods are not competitive, especially compared with modern baselines such as DeiT3 (Touvron et al., 2022), which in essence are better trained ViTs. For example, the best performing SPViT-DeiT-B with distillation failed to outperform a better trained DeiT3-S baseline in both accuracy, model size, and MACs. It remains to be seen whether these methods work with the improved baseline models. In contrast, the process of choosing $T$ for lossless compression with Little-Big in Section 4.2 will yield $T = 0$ which automatically suggests replacement of the Big with the more performant Little model. Adaptive compute may still be an interesting direction, however popular models such as A-ViT (Yin et al., 2022) also fail to match the performance of better trained baseline models or show that their adaptive models are really better than simply scaling down the model moderately to match the MACs of the adaptive counterparts.

## 5 DISCUSSION

A large corpus of literature in modern computer vision (Cai et al., 2023; Dosovitskiy et al., 2020; He et al., 2016; Krizhevsky et al., 2012; Szegedy et al., 2015; Tan & Le, 2019; Touvron et al., 2022) has followed the norm of developing **single-pass** solutions trained **end-to-end** for a wide variety of vision tasks ranging from image/video classification to dense prediction.

**Table 3: Little-Big outperforms a variety of pruning methods and adaptive compute models (†).** Models in blue denote the Little model is from a different model family as the Big model. Thresholds in red indicate the Little model can simply replace the Big model, achieving model compression and better accuracy. "+disl." denotes additional distillation training.

| Model | Method | ImageNet-1K | | GMACs | |
|---|---|---|---|---|---|
| | | Accuracy (%) | $\Delta$ | Remaining | $\Delta$ |
| DeiT-S-224 Touvron et al. (2021) | Baseline | 79.81 | +0.9 | 4.6 | —— |
| | DynamicViT-DeiT-S† Rao et al. (2021) | 78.3 | −0.6 | 3.4 | −26% |
| | A-ViT-S† Yin et al. (2022) | 78.6 | −0.3 | 3.6 | −22% |
| | A-ViT-S† + disl. Yin et al. (2022) | 80.7 | +1.8 | 3.6 | −22% |
| | eTPS-DeiT-S Wei et al. (2023) | 79.7 | +0.8 | 3.0 | −35% |
| | dTPS-DeiT-S Wei et al. (2023) | 80.1 | +1.2 | 3.0 | −35% |
| | SPViT-DeiT-S He et al. (2024) | 78.3 | −0.6 | 3.3 | −28% |
| | SPViT-DeiT-S + disl. He et al. (2024) | 80.3 | +1.4 | 3.3 | −28% |
| | +EfficientViT-B1-224$_{(T=0.44)}$ | 79.81 | +0.9 | **1.1** | **−77%** |
| | DeiT3-S Baseline Touvron et al. (2022) | 83.05 | +4.1 | 4.6 | —— |
| | +EfficientViT-B1-224$_{(T=0.54)}$ | **83.11** | **+4.2** | 2.1 | −54% |
| | +EfficientNet-B2-288$_{(T=0.52)}$ | 83.06 | +4.1 | 2.0 | −56% |
| DeiT-B-224 Touvron et al. (2021) | Baseline | 81.81 | —— | 17.6 | —— |
| | DynamicViT-DeiT-B† Rao et al. (2021) | 81.4 | −0.4 | 11.4 | −35% |
| | SCOP Tang et al. (2020) | 79.7 | −2.1 | 10.2 | −42% |
| | UVC Yu et al. (2022b) | 80.57 | −1.24 | 8.0 | −55% |
| | WDPruning Yu et al. (2022a) | 80.76 | −1.05 | 9.9 | −44% |
| | X-Pruner Yu & Xiang (2023) | 81.02 | −0.99 | 8.5 | −52% |
| | eTPS-DeiT-B Wei et al. (2023) | 81.1 | −0.7 | 11.4 | −35% |
| | dTPS-DeiT-B Wei et al. (2023) | 81.2 | −0.6 | 11.4 | −35% |
| | SPViT-DeiT-B He et al. (2024) | 81.5 | −0.4 | 8.4 | −52% |
| | SPViT-DeiT-B + disl. He et al. (2024) | 82.4 | +0.6 | 8.4 | −52% |
| | +DeiT-S $_{(T=0.60)}$ | 81.83 | +0.02 | 9.0 | −49% |
| | DeiT3-B Baseline Touvron et al. (2022) | 85.75 | +3.94 | 17.6 | —— |
| | +DeiT3-S$_{(T=0.60)}$ | 85.75 | +3.94 | 11.4 | −35% |
| | +EfficientViT-B2-288 $_{(T=0.60)}$ | **85.77** | **+3.96** | 7.8 | −56% |
| Swin-S-224 Liu et al. (2021) | Baseline | 83.17 | —— | 8.7 | —— |
| | STEP Li et al. (2021) | 79.6 | −3.6 | 6.3 | −28% |
| | WDPruning Yu et al. (2022a) | 81.8 | −1.4 | 6.3 | −28% |
| | X-Pruner Yu & Xiang (2023) | 82.0 | −1.2 | 6.0 | −31% |
| | SPViT-Swin-S He et al. (2024) | 82.4 | −0.6 | 6.1 | −30% |
| | SPViT-Swin-S + disl. He et al. (2024) | 83.0 | −0.2 | 6.1 | −30% |
| | +Swin-T $_{(T=0.68)}$ | **83.18** | **+0.01** | 7.0 | −20% |
| | +EfficientViT-B2-224 $_{(T=0.58)}$ | **83.18** | **+0.01** | **2.5** | **−71%** |

While many multi-pass test-time augmentations (TTAs) (Shanmugam et al., 2021) that aggregate predictions of several augmented views of the same sample have been developed as a post-hoc add-on to improve performance, these methods go in the opposite direction of Little-Big . Fundamentally, $K$-pass TTAs are *multiplicative* methods that trades $K\times$ inference cost for slight improvement of accuracy. Little-Big is a *subtractive* multi-pass algorithm that relies on a good decomposition of problems and solve each part with the least compute, not unlike Speculative Decoding in languange modeling (Leviathan et al., 2023). What TTA and Little-Big share in common is that both are post-hoc methods that do not require any re-training of the original models. It is actually possible to combine both in the same inference pipeline much like human vision to make predictions adaptively.

A potentially important contributing factor to the sub-optimal performance of popular adaptive compute models such as DynamicViT and A-ViT lies in their end-to-end training protocol. As the number of tokens decrease over depth, the deeper layers effectively "see" less pixels during training. This implicit coupling between the need to reduce compute cost at inference time and at training time due to the end-to-end training protocol may hinder the potential of adaptive methods. In contrast, Little-Big decouples inference and training compute costs, and uses models individually trained on all pixels on full datasets.

In language modeling, recent work such as Hybrid-LLM (Ding et al., 2024) shares the same principle as Little-Big . Hybrid-LLM involves training a probabilistic router that connects a little transformer and a big transformer. Like many other model compression methods, Hybrid-LLM trades accuracy for inference speed. In contrast, the embarrassingly simple routing mechanism of Little-Big involves choosing only one hyperparameter $T$, and achieves lossless speedup without losing accuracy.

**Limitations** It is worth noting that, while Little-Big achieves drastic MACs reduction across model families and scales, the speed up is **not free**: since an additional Little model is needed to make the Little-Big pair, the storage overhead is usually a small fraction of the storage requirement of the Big model. However, the storage overhead does not necessarily translate to memory overhead. Although

the example Pytorch pseudo code keeps both Big and Little models in memory at the same time to minimize latency while increasing maximum memory usage, one can alternatively only load one model into the memory at a time so that there is no overhead on maximum memory usage. Batching predictions can further reduce the memory I/O overhead per sample.

**Extensions**    It is possible to extend the same principles of Little-Big to other tasks by re-examining and modifying Equation 1. For example, consider the task of video classification, where the input $x$ is updated as a sequence of images with an added time dimension $t$, $x \in \mathbb{R}^{C \times W \times H \times t}$. The inference cost of a neural video classifier is given by:

$$MACs[F(x)] \approx C_F * H^2 * t * w^2 * l, \tag{6}$$

which makes brute-force upscaling more costly because of the additional $t$ dimension. However, decomposition based on Equations 4 and 5 generalize to this task provided video classifiers are well calibrated and video samples are decomposable by confidence. To demonstrate the generalizability of Little-Big , we apply it to the Swin3D-B video classification model on Kinetics-400. The Swin3D-T + Swin3D-B pair achieves 41% MACs reduction of Swin3D-B while maintaining 79.4% Top-1 accuracy on Kinetics-400. For dense image prediction tasks like semantic segmentation, the prediction $y$ becomes a map, where $y \in [0,1]^{W \times H \times N}$. In principle, one can continue perform decomposition on a per sample basis, but it may be more efficient to perform decomposition on a per pixel or per patch level. In Appendix A.1 we also demonstrate how Little-Big could also be extended to zero-shot multimodal classification, semantic segmentation, multi-label segmentation as well as for text classification

**Additional contextualization of novelty**    As the ML community continues to produce increasingly large models with massive parameter counts, efficiently deploying them has become a significant challenge. Current inference optimization strategies, such as quantization or distillation, often require training new, smaller models or performing operations that result in performance drops. Additionally, these techniques fail to fully leverage the extensive ecosystem of models that is readily available (e.g., the EfficientNet family or user-submitted models on platforms like Torch or HuggingFace Hub).

In this context, we believe that our study on speeding up large models using smaller models is an important direction of work. Though the proposed protocol is simple to implement, our work holds significant practical value since (i) our approach completely **post-hoc**, requiring no re-training; (ii) our approach is **entirely model- and architecture-agnostic**, allowing seamless integration with a variety of models; and ours is the first work to systematically study and benchmark its utility in improving inference efficiency across a variety of tasks and model architectures.

By leveraging the growing diversity of pre-existing models across frameworks, Little-Big enables users to mix and match architectures (e.g., pairing models from different families such as EfficientNet and ViTs, BERTs and T5s). This flexibility ensures that one can adopt state-of-the-art models without the burden of additional training or specialized pipelines, and obtain significant compute savings and latency reductions without sacrificing accuracy. We argue that the framework's simplicity, adaptability, and compatibility with existing models make it a highly practical solution for real-world use cases.

## 6    CONCLUSION

In summary, we investigate how scaled-up models help base models correct their mistakes, show that the former preferentially help with samples with low confidence predictions. Inspired by that, we propose a simple two-pass Little-Big algorithm that selectively pass "difficult" samples to large models, achieving drastic MACs reduction of up to $80\%$ without sacrificing accuracy for a wide range of model families and sizes.

The inefficiency in model scale-up and the effectiveness of Little-Big in compressing the scaled-up models are two sides of the same coin, with effective decomposition of samples in a dataset being the key to connecting the two sides. Despite being embarrassingly simple and surprisingly effective, Little-Big is considered as a lower bound on how much a model can be compressed without losing accuracy. More sophisticated ways to decompose the data distribution and better ways to use the Little models output are promising directions to further improve the performance of such subtractive multi-pass algorithms.

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

# A APPENDIX

## A.1 EXTENSIONS AND ADDITIONAL INSIGHTS

### A.1.1 EXTENDING TO ZERO-SHOT CLASSIFICATION USING MULTIMODAL MODELS

To evaluate the generality of the Little-Big paradigm, we apply it to zero-shot classification using OpenAI's pre-trained CLIP models, specifically with ViT-B-32 and ViT-L-14 as the little and big models respectively. We conducted experiments on four diverse datasets: Flowers-102 (Nilsback & Zisserman, 2008), Food-101 (Bossard et al., 2014), SUN-397 (Xiao et al., 2010), and DTD (Cimpoi et al., 2014). We follow the same protocol as outlined in Section 4.1 with a key difference in how logits are obtained. In CLIP models, logits are computed as the cosine similarity between the image embedding and the text embeddings of class labels. We construct the prompts for each class using the prompt templates outlined in the OpenAI's CLIP repository. We then apply softmax over these logits to derive class probabilities and using the little model, we establish a threshold $T$ based on 20% of the data (validation split) and measure the performance of the Little-Big framework on the remaining 80%.

Table 4 summarizes the results. For each dataset, we report the top-1 accuracy achieved by the Little model, the Big model, and the performance of Little-Big and Little-Big Ensemble approaches.

**Table 4:** Zero-shot classification results using CLIP models on Flowers-102, Food-101, SUN-397, and DTD datasets. The Little-Big approach effectively balances accuracy and computational costs.

| Dataset | Little Model Acc (%) | Big Model Acc (%) | Little-Big Acc (%) | Little-Big $\Delta$ GMACs (%) | Little-Big Ensemble Acc (%) | Little-Big Ensemble $\Delta$ GMACs (%) |
|---|---|---|---|---|---|---|
| SUN-397 | 54.09 | 58.50 | 58.44 | -49.91 | 58.68 | -63.11 |
| Food-101 | 78.42 | 89.79 | 89.41 | -45.16 | 89.42 | -47.09 |
| DTD | 31.86 | 37.44 | 37.44 | -31.20 | 37.6 | -37.45 |
| Flowers-102 | 53.29 | 66.35 | 65.31 | -33.49 | 65.34 | -33.40 |

These results demonstrate that the Little-Big framework maintains the high accuracy of the big model while significantly reducing the computational cost, confirming its adaptability to multimodal zero-shot tasks.

### A.1.2 EXTENDING TO MULTI-LABEL CLASSIFICATION

To evaluate the applicability of the Little-Big framework in multi-label classification, we conducted experiments using the CelebA dataset (Liu et al., 2015), which contains 40 attributes (classes). In this setup, we used Vision Transformers (ViTs) pretrained on ImageNet: with `ViT-B-32` as the "Little" model, and `ViT-L-14` as the "Big" model. With the backbones frozen, we trained a classification head— comprising of two linear layers with ReLU activation and dropout on the CelebA dataset.

Since the multi-label classification is typically posed as $k$ binary predictions for each sample, where $k$ is the number of classes, we obtain $k$ logits per sample. A sigmoid function is then applied to obtain the probabilities for each class. We compute the confidence for a class as confidence $= |\text{prob} - 0.5|$ measuring the distance from the decision boundary. To obtain an *aggregated confidence* for an image, the confidences across all $k$ classes are averaged.

Using this aggregated confidence from the predictions of the Little model, we determine a threshold $T$ on a validation subset and pass the images whose aggregated confidence is below $T$ to the big model. Through the results, reported in Table 5, we demonstrate the effectiveness of the Little-Big framework for multi-label classification, with minimal loss in F1-score while significantly reducing computational costs.

**Table 5:** Multi-label classification results using `ViT-B-32` and `ViT-L-14` models on the CelebA dataset. Little-Big speeds up the big model significantly with a small drop in performance.

| Little Model F1 (%) | Big Model F1 (%) | Little-Big F1 (%) | $\Delta$ GMACs (%) |
|---|---|---|---|
| 60.44 | 64.31 | 63.28 | -40.10 |

### A.1.3 Extending Little-Big to Semantic Segmentation

To evaluate the Little-Big framework in the context of semantic segmentation, we conducted experiments using DeepLabv3 (Chen, 2017) with MobileNet (Howard, 2017) as the "Little" model and FCN (Long et al., 2015) with ResNet-101 as the "Big" model. Both models were sourced from the PyTorch repository and pretrained on a subset of the COCO dataset (Lin et al., 2014) that includes class names overlapping with Pascal VOC (Hoiem et al., 2009). These models were not fine-tuned further and evaluation was performed on the validation set.

In this setup, we first generated superpixels for each input image using the SLIC algorithm (Achanta et al., 2012), producing compact regions that preserve spatial information. The image was then passed through the Little model to obtain class-wise probabilities for each pixel. For each SLIC region, we computed a region-level confidence by averaging the maximum softmax probability across all pixels in the region. These region-level confidences were further aggregated across the image to compute an *image-level aggregated confidence*, summarizing the overall certainty of the Little model's predictions.

Based on the image-level aggregated confidence, a threshold $T$ was applied and images with confidence below $T$ were passed to the Big model for further evaluation. This two-stage approach leverages the efficiency of the little model while ensuring high-quality predictions via the big model when necessary. The results, detailed in Table 6, demonstrate that the Little-Big framework effectively balances computational efficiency with segmentation accuracy in this setting.

**Table 6:** Semantic segmentation results using the Little-Big framework.

| Little Model mIOU (%) | Big Model mIOU (%) | Little-Big mIOU (%) | $\triangle$ GMACs (%) |
|---|---|---|---|
| 60.3 | 63.7 | 63.0 | -39.2 |

### A.1.4 Extending Little-Big to NLP Classification

Finally, we extend the framework to NLP tasks by conducting experiments on the IMDB movie review sentiment classification task (Maas et al., 2011). In this setup, DistilBERT (Sanh, 2019) served as the "Little" model, and GPT2 (Radford et al., 2019) was used as the "Big" model. Both models were sourced from the HuggingFace hub and fine-tuned on this dataset. We employ the same approach as outlined in Section 4.1. The results detailed in Table 7 clearly evidence the effectiveness of the approach to even non-vision classification tasks by achieving significant computational savings without losing performance.

**Table 7:** IMDB sentiment (binary) classification using Distill-Bert and GPT2 as Little and Big models.

| Little Model Acc (%) | Big Model ACC (%) | Little-Big ACC (%) | $\triangle$ GMACs (%) |
|---|---|---|---|
| 92.8 | 93.5 | 93.51 | -58.75 |

## A.2 Additional Experimental Details

**Implementation** The three datasets used in this work, ImageNet-1K, ImageNet-ReaL, and ImageNet-V2 are released with ImageNet license https://www.image-net.org/download.php, Apache-2.0 license, and MIT license, respectively. All models are implemented in Pytorch Paszke et al. (2019). EfficientNet Tan & Le (2019), Swin Liu et al. (2021), ConvNext Liu et al. (2022), and ViT Dosovitskiy et al. (2020) checkpoints are loaded from public Torchvision pretrained models with BSD-3 license. Pretrained models including EfficientViT Cai et al. (2023), DeiT Touvron et al. (2021), DeiT3 Touvron et al. (2022) are accessed on Timm Wightman (2019) under Apache-2.0 license. The official code and weight of InternImage models Wang et al. (2022a) are accessed under MIT licence.

Model inference are conducted on an NVIDIA RTX 3090 with 24 GB vRAM, taking  1 minute to  4 hours to finsh evaluation on the ImageNet-1K validation set. The main results are based on

measurements of single pretrained models. To estimate the error bar on single model accuracy, we train a small EfficientNet-B0 on ImageNet-1K from scratch for 300 epochs on 32 NVIDIA V100 GPUs with cosine learning rate decay Loshchilov & Hutter (2016), TrivialAugment Müller & Hutter (2021), input resolution of 224, batch size of 2048, AdamW Loshchilov & Hutter (2017) with initial learning rate of 0.003, and weight decay of 0.05. Three models are independently trained from scratch and achieve top-1 accuracies of 76.76%, 76.43%, 76.56%, with a standard deviation of 0.14%.

As shown in Section 4.2, the dataset used in choosing $T$ can affect the optimal $T$ as well as relative MACs reduction. By choosing $T$ over ImageNet-1K, Real, V2 for the same model pair, we estimate MACs reduction reported in Tables 2 and 3 to be $\sim 2\%$, which is an order of magnitude smaller than the effect size of $50\%$ throughout the paper, validating the statistical significance of our speedup.

**Table 8:** To simulate errors stemming from noise in determining the optimal $T$, we choose $T$ for each Little-Big pair on ImageNet-1K, Real, and V2 and compute mean and standard deviation of accuracy change and MACs reduction on ImageNet-1K.

| Choose $T$ on: | | IN-1K | | ReaL | | V2 | | Mean | | SD | |
|---|---|---|---|---|---|---|---|---|---|---|---|
| | | $\Delta Acc_{1K}$ | $\Delta GMACs_{1K}$ | $\Delta Acc_{1K}$ | $\Delta GMACs_{1K}$ | $\Delta Acc_{1K}$ | $\Delta GMACs_{IN}$ | $\Delta Acc_{1K}$ | $\Delta GMACs_{IN}$ | $\Delta Acc_{1K}$ | $\Delta GMACs_{IN}$ |
| Big Model | Little Model | | | | | | | | | | |
| | B6-528 | +0.02 | −47% | −0.01 | −48% | −0.11 | −49% | −0.03 | −48% | 0.06 | 1% |
| | B5-456 | +0.01 | −65% | +0.09 | −62% | −0.20 | −68% | −0.03 | −65% | 0.12 | 2% |
| | B4-380 | +0.01 | −81% | +0.05 | −80% | +0.09 | −78% | +0.05 | −80% | 0.03 | 1% |
| EfficientNet-B7-600 | B3-300 | +0.02 | −61% | −0.02 | −65% | −0.01 | −69% | −0.00 | −65% | 0.02 | 3% |
| | B2-288 | +0.00 | −59% | +0.01 | −53% | −0.16 | −67% | −0.05 | −60% | 0.08 | 6% |
| | B1-240 | +0.01 | −13% | −0.01 | −18% | −0.12 | −51% | −0.04 | −27% | 0.06 | 17% |
| | B0-224 | +0.01 | −25% | +0.01 | −25% | −0.03 | −44% | −0.01 | −31% | 0.02 | 9% |

As discussed in Section 2.3, the roles of Big and Little in the Little-Big pair are **relative**. For example, DeiT3-L-384 performs well as Little model, efficiently speeding up the 3B-parameter InternImage-G-512 Wang et al. (2022b) and 600M-parameter ViT-H-14-518 by 62% and 80%, respectively. However, it does not imply that DeiT3-L-384 itself cannot act as the Big model and be sped up by an even smaller model. In fact, Table 2 shows that it can be sped up by the EfficientViT-L2-288 by 71%. Similarly, EfficientNet-B4 can serve as a strong Little model for compressing EfficientNet-B7, as well as a Big model to be compressed by the smaller EfficientNet-B2. This enables the generalization of Little-Big to a $K$-pass framework where $K > 1$ with additional performance gain.

One can extend Little-Big to a 3-pass Tiny-Little-Big model $G_{F_{Tiny}, F_{Little}, F_{Big}}(x, T_1, T_2)$ or shortly $G(x, T_1, T_2)$ for simplicity:

$$
G_{F_{Tiny}, F_{Little}, F_{Big}}(x, T_1, T_2) = \begin{cases} F_{Tiny}(x), \text{if } max(F_{Tiny}(x)) \geq T_1 \\ F_{Little}(x), \text{if } max(F_{Tiny}(x)) < T_1, max(F_{Little}(x)) \geq T_2 \\ F_{Big}(x). \end{cases}
$$

(7)

A 3-pass model $G_{B2,B4,B7}(x, T_1 = 0.74, T_2 = 0.26))$ further compresses EfficientNet-B7, achieving 85% MACs reduction while preserving the Big model performance.

### A.3 LITTLE-BIG ENSEMBLE

A more sophisticated way to construct Little-Big pairs is to combine the softmax output of the little and big models for hard examples. $G_{F_{Little}, F_{Big}}(x, T, w)$ or shortly $G(x, T, w)$ for simplicity:

$$
G_{F_{Little}, F_{Big}}(x, T, w) = \begin{cases} F_{Little}(x), \text{if } max(F_{Little}(x)) \geq T \\ (1 - w) * F_{Little}(x) + w * F_{Big}(x). \end{cases} \qquad \textbf{Little-Big Ensemble} \quad (8)
$$

where weight $w$ controls the relative weights of the two models. the softmax score $F_{Little}(x)$ is used for model ensembling. In the case of $w = 1$, this reverts to the base Little-Big prescribed by Equation 4.

918
919
920
921
922
923
924
925
926
927
928
929
930
931
932
933
934
935
936
937
938
939
940
941
942
943
944
945
946
947
948
949
950
951
952
953
954
955
956

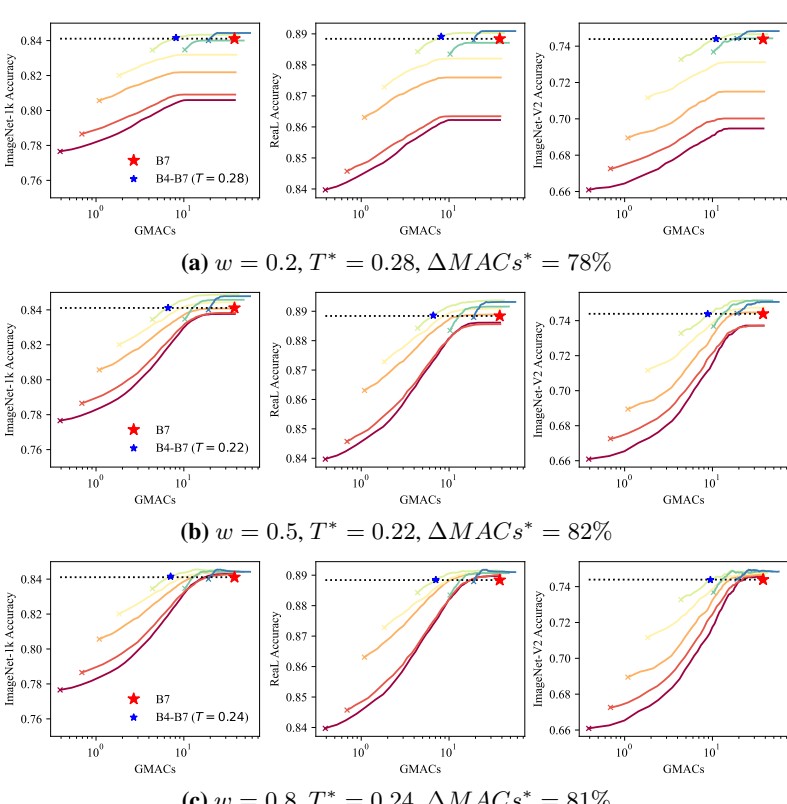

**(a)** $w = 0.2, T^* = 0.28, \Delta MACs^* = 78\%$

**(b)** $w = 0.5, T^* = 0.22, \Delta MACs^* = 82\%$

**(c)** $w = 0.8, T^* = 0.24, \Delta MACs^* = 81\%$

957
958
959
960
961
962
963
964
965
966
967
968
969
970
971

**Figure 4:** Speeding up EfficientNet-B7 with the EfficientNet family and Little-Big Ensemble (Equation 8).

In Figure 5, EfficientNet-B7 is compressed without loss of accuracy by Little-Big Ensemble (Equation 8) using EfficientNet-B0-B6. Setting $w = \{0.2, 0.5, 0.8\}$ yield $T^* = \{0.28, 0.22, 0.24\}$ and consequently MACs reductions $\Delta MACs^* = \{78\%, 82\%, 81\%\}$, similar to $T = 0.24$ and $\Delta MACs = 81\%$ achieved by the simple Little-Big (Equation 4).

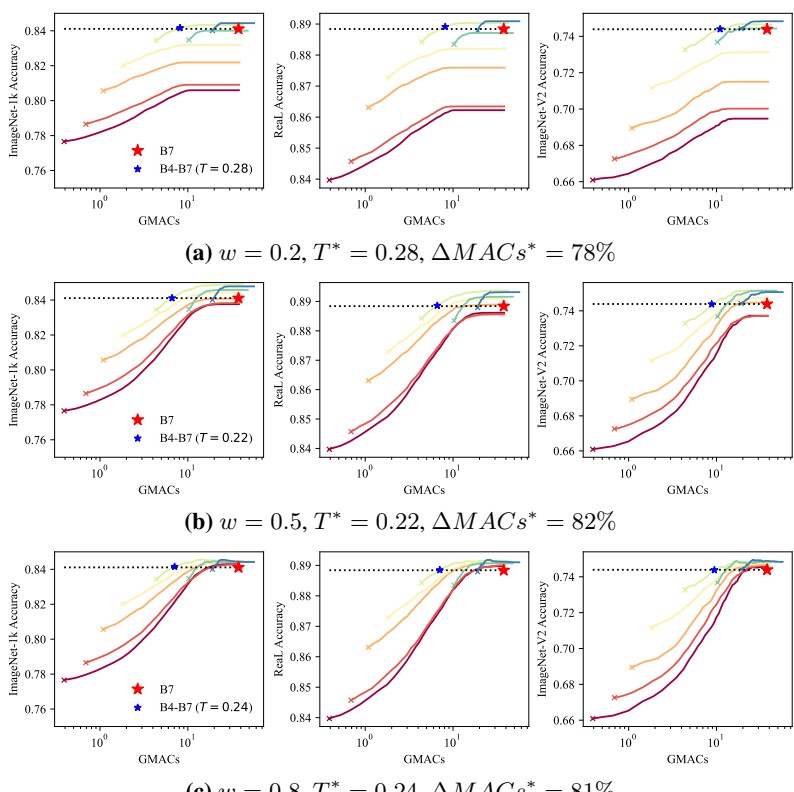

**(a)** $w = 0.2$, $T^* = 0.28$, $\Delta MACs^* = 78\%$

**(b)** $w = 0.5$, $T^* = 0.22$, $\Delta MACs^* = 82\%$

**(c)** $w = 0.8$, $T^* = 0.24$, $\Delta MACs^* = 81\%$

**Figure 5:** Speeding up EfficientNet-B7 with the EfficientNet family and Little-Big Ensemble (Equation 8).

### A.4 BENCHMARKING THROUGHPUT AND LATENCY

Average throughput and latency are measured on an Nvidia RTX3090 on the ImageNet-1K validation set with 50000 samples. In Table 9, we consider a memory constrained use case where the max memory use of Little-Big pair is no larger than that of using the Big model alone. This requires the Little model being unloaded from memory after the first passs before the big model is loaded for the second pass. Without the memory constraint, Little-Big can achieve better throughput and latency improvement than resported in Table 9.

**Table 9:** We use the largest power of 2 batch size that fits in the vRAM for each model, throughput and latency include amortized model loading time. Compared with EfficientNet-B7, EfficientNet-B4+B7 pair achieves throughput improvement by 339% and latency reduction by 77%.

| | | ImageNet-1K | | Throughput | | Latency | |
|---|---|---|---|---|---|---|---|
| Big Model | Little Model | $\Delta Acc$ | $\Delta GMACs$ | samples/s | $\Delta$ | ms | $\Delta$ |
| | None | $--$ | $--$ | 35.5 | $--$ | 28.2 | $--$ |
| | B0-224 | $+0.01$ | $-25\%$ | 54.0 | $+52\%$ | 18.5 | $-34\%$ |
| | B1-240 | $+0.01$ | $-13\%$ | 39.9 | $+12\%$ | 25.1 | $-11\%$ |
| | B2-288 | $+0.00$ | $-59\%$ | 76.6 | $+116\%$ | 13.1 | $-54\%$ |
| EfficientNet-B7-600 | B3-300 | $+0.02$ | $-61\%$ | 86.4 | $+144\%$ | 11.6 | $-59\%$ |
| | B4-380 | $+0.01$ | $-81\%$ | 155.2 | $+338\%$ | 6.4 | $-77\%$ |
| | B5-456 | $+0.01$ | $-65\%$ | 97.1 | $+174\%$ | 10.3 | $-64\%$ |
| | B6-528 | $+0.02$ | $-47\%$ | 62.4 | $+76\%$ | 16.0 | $-43\%$ |

To demonstrate the generalizability of Little-Big, we apply Little-Big to the Swin3D-B video classification model on Kinetics-400. The Swin3D-T + Swin3D-B pair achieves 41% MACs reduction of Swin3D-B while maintaining 79.4% Top-1 accuracy on Kinetics-400.

## A.5 ADDITIONAL DATA

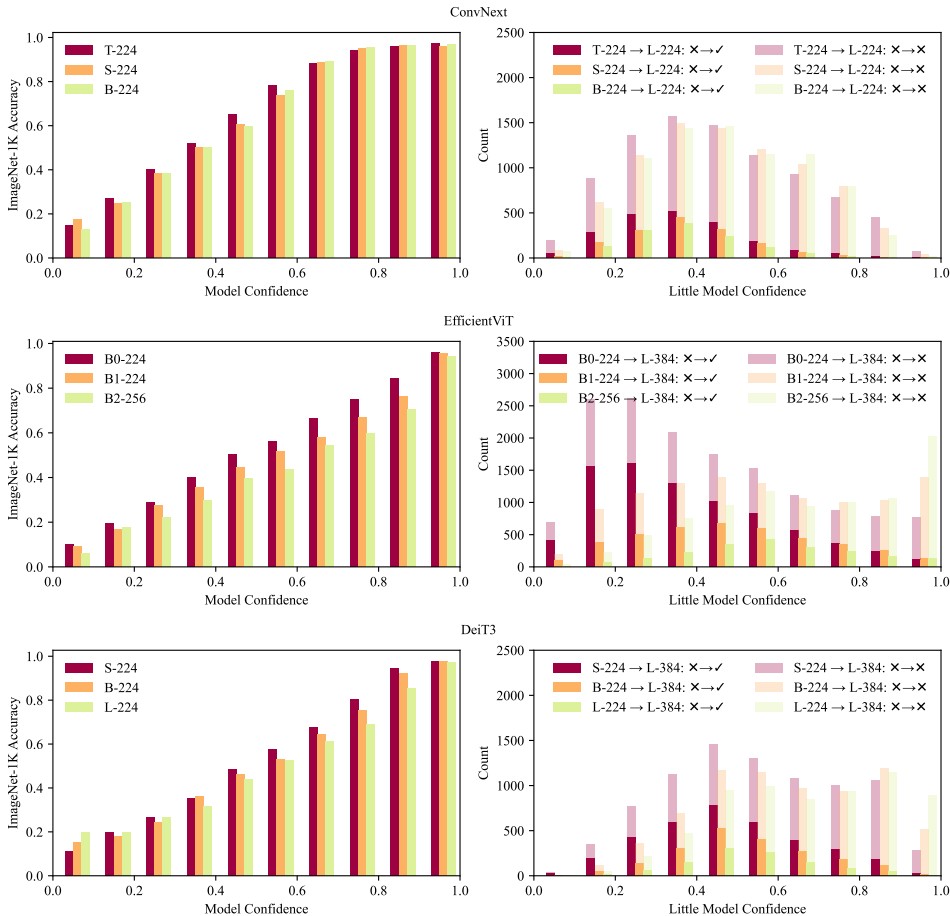

**Figure 6:** More examples with ConvNext, EfficientViT, and DeiT3. Prediction confidence of individual models correspond well with prediction accuracy (left), which allows us to approximate a **"difficulty"** axis with prediction confidence. Breaking down the mistakes of little models by difficulty, we find that big models disproportionally "correct" mistakes that are difficult to the little models.

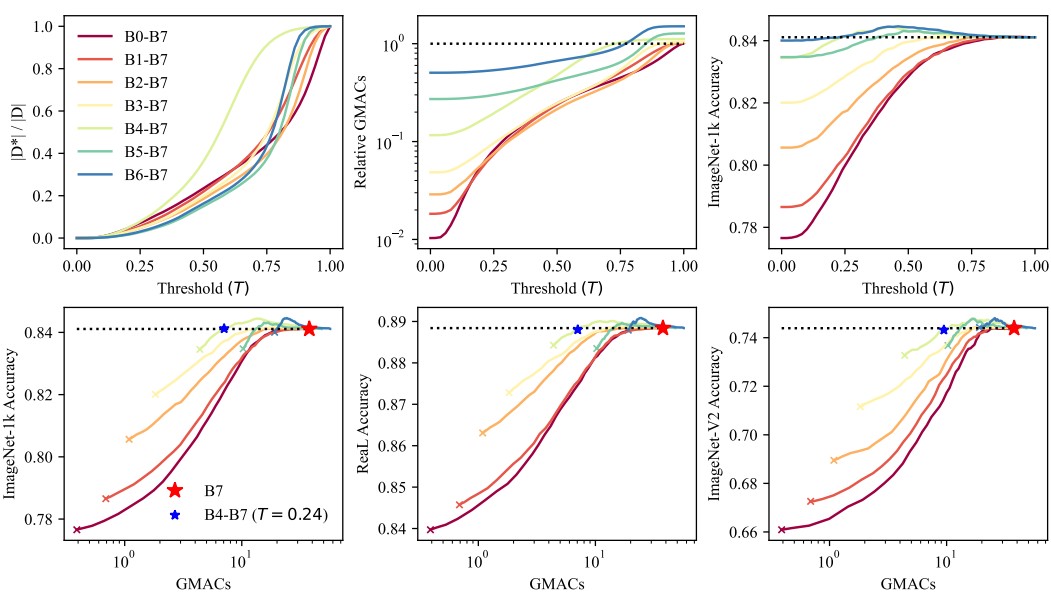

**Figure 7:** Speeding up EfficientNet-B7 with the EfficientNet family. Different from Figure 3 where $T$ is chosen on ImageNet-1K, the optimal pair (blue star) is chosen on the smaller V2 set. This yields a similar optimal $T = 0.28$ achieving 78% of MACs reduction.

