# OpenReview forum: "Speeding Up Image Classifiers with Little Companions"
_ICLR.cc/2025/Conference — ICLR 2025 Conference Withdrawn Submission_

### Official Review · Reviewer_ekkS · 2024-10-18

**Soundness:** 2
**Presentation:** 4
**Contribution:** 2
**Rating:** 6
**Confidence:** 4

**Summary:**

The paper discusses improving the speed of visual recognition systems using a "Little-Big" model setup. The Little model is a smaller architecture that processes examples first. If the confidence is below a predefined threshold, the sample is reprocessed by a "Big" model. This simple setup improves the speed of visual recognition systems on ImageNet, ImageNetv2, and ImageNetReal significantly (without loss in accuracy). The authors experiment with both CNNs and Transformer models. They also study the fraction of examples processed by the Little and the Big network, showing the accuracy as a function of threshold.

**Strengths:**

- The paper studies an important topic - efficiency of visual recognition models.
- The speedup claimed by the paper is substantial. At a fixed accuracy, their method improves speed by 30%-80% (Figure 1).
- The paper is written clearly and is easily understandable. The investigation presented studies the natural questions that arise with threshold tuning for the Little model's confidence. Figure 3 clearly demonstrates how the accuracy and efficiency change as a function of the threshold.
- The paper accounts for generalization across different datasets by fixing the threshold on ImageNet and analyzing results on ImageNetv2 and ImageNetReal. This is an important aspect of the investigation, as choosing the threshold based on the validation set can create bias.

**Weaknesses:**

My main concern is the comparison with prior art. As line 437-439 states, "even with tricks that effectively retrained models, many pruning methods are not competitive... with modern baselines... which in essence are better trained ViTs". It seems that the Little-Big method is evaluated using modern architectures and training recipes, whereas other baselines (pruning, etc.) are using older architectures or training recipes. I'm worried that the gains of this method are primarily attributable to the use of newer architectures or training recipes. A fair comparison would use the same architectures as previous works.
- For example, Table 3 shows a datapoint with T=0, meaning the Big architecture is never used.
- Additionally, the baseline architecture in Table 3 ("Our Baseline") is significantly more accurate than previous work's baseline (Yin et al.).

The choice of "Little" network seems arbitrary in some cases. In Table 2, EfficientNet-B2-288 uses EfficientViT as a little network, but most other EfficientNet variants do not. And EfficientNet-B2-288 is not used with EfficientNet-B1, but most other variants are. I have similar thoughts on most of the rows in Table 2. Can you please justify the choice of Little architecture?

Line 132: Equation 2 should have a reference, and there should be some more specific qualifications as to what types of models this equation applies to. Similarly, the characteristic width w_j is not well defined and doesn't have a reference.

Line 150: I recommend also discussing quantization briefly here.

Line 172-173: "ingest gigabits/s of raw visual information and compress it to tens of bits/s" <- this needs a reference

In Table 3, it would help the reader if you mark the baselines by their general approach (e.g. which ones are pruning, etc.).

Line 202: "confidence > 0.5-0.7" <- what does this mean? How can a confidence be greater than a range? Did you instead mean "0.5 < confidence < 0.7"?

**Questions:**

My main suggestion is regarding the fairness of comparisons with baselines (mentioned above).

I also would like to understand the justification for the choice of Little model (mentioned above).

---

> ### Author Response · Authors · 2024-11-22
>
> **Quantization**
>
> Quantization is indeed relevant for model compression. While the focus of Little-Big is on inference-time compute savings without retraining, we acknowledge that quantization could further complement Little-Big. We have aded a brief discussion in Section 2.3, positioning quantization as a complementary method that can be integrated with Little-Big for additional gains.
>
> **Additional Experiments**
> We also want to request the reviewer to please check our summary response that reports our findings in extending Little-Big to novel tasks and domains including semantic segmentation, zero-shot classificaiton, multi-label classifcaition as well as extensions to text classification tasks. Details of these new added experiments can be found in Appendix A.1 of the revised paper.
>
>
>
>
> **Fair comparison with Baselines**
>
> Thank you for this question.  In the A-ViT paper by Yin et al., the authors report the baseline DeiT performance as 78.9%, whereas the original DeiT paper reports a performance of 79.81% for the same model. Additionally, we confirmed the 79.81% performance through the publicly released DeiT-S checkpoint. Since it was not immediately clear which of these values to report, and to ensure fairness when comparing against other methods, we initially included both. However, we acknowledge that this choice could cause confusion for the reader. To address this, we have revised the paper to only include the 79.81% baseline from the original DeiT paper for clarity and consistency.
>
> We also made similar changes in Table-3 with  DeiT-B architecture to improve readability. Little-Big pairs with T=0.0 indeed suggest that a big model can be replaced by a small model. In the original table, we included such pairs mainly to highlight that an effective but often neglected way of model compression is to train a smaller model with a better training recipe (e.g. DeiT3-S outperforms DeiT-B). But we acknowledge that it may cause confusion and removed such pairs with T=0.0 from Table 3.
>
> **Choice of Little Network**
>
> Thank you for pointing this out. We want to highlight that the choice of the "Little" network is, in fact, user-dependent and flexible. Little-Big imposes no strict requirement on which model to use as the Little model, and it is entirely architecture-agnostic. Users can select a Little model based on their specific constraints, such as available compute resources, latency requirements, or model compatibility with the Big model.
>
> **Typographical Error**
>
> Additionally, the phrasing “confidence > 0.5-0.7” was indeed an error and instead It should have been  confidence > 0.5. To clarify, our intent was to convey that confidence in the models we examined inversely correlates with hardness. Specifically, from the distribution of correctable mistakes, we observed that most of these samples exhibit lower confidence.  Thus, this observation directly motivates our approach of thresholding over confidence: when the confidence of a prediction falls below the threshold, the Big model is invoked to handle the sample. We thank you for highlighting this issue, and we have updated the phrasing and explanation in the revised paper for clarity and accuracy.
>
> **Additional References**
> We have added the following references to the line "human vision ingests gigabits/s and compresses to tens of bits"  in the revised paper.
> Soto, Florentina, Jen-Chun Hsiang, Rithwick Rajagopal, Kisha Piggott, George J. Harocopos, Steven M. Couch, Philip Custer, Josh L. Morgan, and Daniel Kerschensteiner. "Efficient coding by midget and parasol ganglion cells in the human retina." Neuron 107, no. 4 (2020): 656-666.
>
> Zheng, Jieyu, and Markus Meister. "The Unbearable Slowness of Being." arXiv preprint arXiv:2408.10234 (2024).
>
> We hope our responses answer your concerns.

---

> ### Comment · Reviewer_ekkS · 2024-11-25
> **Response to Rebuttal**
>
> *Fair Comparisons With Baselines*
>
> Yes, I see that DeiT accuracy has increased over time due to improved training recipes. I’m worried that your dominance over other pruning baselines is for the same reason. If these baselines had the benefit of an improved recipe, would they be improved as well?
>
> To put it in your own words, as line 415-417 states, "even with tricks that effectively retrained models, many pruning methods are not competitive... with modern baselines... which in essence are better trained ViTs". This seems like an issue, since we don’t know if your method is better since we don’t know how these other methods perform with modern recipes.
>
> *Choice of Little Network*
>
> I’m aware that there’s no restriction. That’s what makes me wonder **why you choose to show only specific combinations of big and little network**, rather than all of them.
>
> In Table 2, EfficientNet-B2-288 uses EfficientViT as a little network, but most other EfficientNet variants do not. And EfficientNet-B2-288 is not used with EfficientNet-B1, but most other variants are. Unless there’s consistency or an obvious pattern in the choices made, the results look cherry-picked. Can you please explain the motivation behind the choice of combinations shown?

---

> > ### Author Response · Authors · 2024-11-29
> > **Response - Comprehensive Evaluation**
> >
> > We thank you for the feedback.
> >
> > **Exhaustive Evaluation of All Possible Model Pairings** In our original submission, we showcased only a subset of combinations in the interest of space, choosing representative examples. However, we understand the concern regarding the consistency and completeness of the choices presented.
> >
> > To address the concerns raised, we have now evaluated all possible little-big pairs among the 63 models, resulting in a comprehensive evaluation of 1,953 combinations (63 choose 2). These results are publicly available and can be explored interactively at https://littlebigpaper.streamlit.app/. This platform allows users to select any big and little model from dropdown menus and view the detailed results for their chosen combination.
> >
> > We answer the specific question about use of EfficientViT as Little model for  Efficient models other than EfficientNet-B2-288 with our new exhaustive list of results. **We observe that with no loss of accuracy- EfficientViT speeds up Efficinet-Net-B3 by 31.35%, EfficinetNet-B4 by 37.8%,  EfficinetNet-B5 by 29%, EfficinetNet-B6 by 34.346% and EfficinetNet-B7 by 36.3%**
> >
> > Through these results, we want to emphasize that our intent with the initially presented combinations was purely demonstrative and not to cherry-pick results. The expanded evaluation includes all 1,953 combinations, even those where loss-less speedup was not possible in which case, we explicitly indicated in the interactive tool. For example Efficient-Vit-B3-224 and Resnet18-v1.

---

> ### Comment · Reviewer_ekkS · 2024-12-01
> **Updated Score**
>
> I increased my score 1 grade, as the authors have added a lot of empirical evaluation. As raised by another review, I do not think the novelty is strong, but I do think the empirical analysis is useful for understanding the relative accuracies of models and their behavior when combined in terms of efficiency/accuracy.

---

### Official Review · Reviewer_j8uk · 2024-10-24

**Soundness:** 3
**Presentation:** 3
**Contribution:** 2
**Rating:** 3
**Confidence:** 3

**Summary:**

This paper proposes a technique called the Little-Big algorithm. It combines a small and large pre-trained model to improve the trade-off between cost and accuracy. It first applies the small model to a given sample. When the confidence is high, the prediction is returned. Otherwise, the large model is applied, and its prediction is used.

In experiments, the authors focused on the ImageNet-1K image classification task. They demonstrated that the proposed method boosts efficiency for various pairs of models.

**Strengths:**

S1) The proposed method is practical. It is easy to implement and does not require any modification or additional training of existing models.

S2) It is widely applicable. For any classification problem, it’s readily available. We could apply it to other tasks as well if we could come up with confidence estimation methods for them.

S3) Extensive experimental results show that the proposed method is robustly performing well in the ImageNet classification task.

**Weaknesses:**

W1) The proposed approach lacks novelty. The idea of using multiple models with different cost-accuracy tradeoffs is highly common, to name a few, such as speculative decoding for language models and cascade ranking systems for recommendation and information retrieval.

W2) The experiments are weak. All the experiments are about ImageNet-1k image classification task, so it is quite uncertain whether this method works well for other tasks as well.

**Questions:**

Q1) Can you develop better ways to explain your idea’s novelty, or do you have any ideas to further enhance the novelty?

Q2) Can this idea be applied to language models as well? Particularly, I'm interested to see how it compares with speculative decoding.

---

> ### Author Response · Authors · 2024-11-22
>
> **Expanded Contextualization of Novelty**
>
> As the ML community continues to produce increasingly large models with massive parameter counts, efficiently deploying them has become a significant challenge. Current inference optimization strategies, such as quantization or distillation, often require training new, smaller models or performing operations that result in performance drops. Additionally, these techniques fail to fully leverage the extensive ecosystem of models that is readily available (e.g., the EfficientNet family or user-submitted models on platforms like Torch or HuggingFace Hub).
>
> In this context, we believe that our study on speeding up large models using smaller models is an important direction of work. Though the proposed protocol is simple to implement, our work holds significant practical value since
> - our approach completely **post-hoc**, requiring no re-training.
> - our approach is entirely **model- and architecture-agnostic**, allowing seamless integration with a variety of models.
> - ours is the first work to systematically study and benchmark its utility in improving inference efficiency across a variety of tasks and model architectures.
>
> By leveraging the growing diversity of pre-existing models across frameworks, Little-Big enables users to mix and match architectures (e.g., pairing models from different families such as EfficientNet and ViTs, BERTs and T5s). This flexibility ensures that one can adopt state-of-the-art models without the burden of additional training or specialized pipelines, and obtain significant compute savings and latency reductions without sacrificing accuracy. We argue that the framework's simplicity, adaptability, and compatibility with existing models make it a highly practical solution for real-world use cases.
>
> **Comparison to speculative decoding and cascaded systems**
>
> Speculative decoding (SpD) has been mainly developed for autoregressive sequence-to-sequence generative models where an output token is then used as a part of the input to the next inference step. With drafts from the smaller model, SpD decreases the large model latency by better utilizing the parallel computing capacity of modern hardware like GPUs, it actually increases the number of Flops. Little-big on the other hand operates in the predictive modeling domain and provides a net Flops reduction together with latency and throughput improvement.
>
> While Little-Big indeed shares similarity with cascading strategies, these methods rely on task-specific architectures  requiring training or fine-tuning at each stage  for sequential re-ranking, Little-Big is a general protocol applicable across domains (e.g., vision, NLP, multimodal tasks) without task-specific modifications.
>
> **Additional Experiments**
>
> We thank you for this question. We want to first point out that in the main paper, we go beyond imagenet classification task and also consider the video-classification task (lines 485-497). Furthermore, we have now extended and studied Little-Big in four additional tasks. We refer the reviewer our summary response where we summarize our findings on extending our approach to various other tasks including text classification, semantic segmentation, multi-label classifiation and zero-shot classification tasks on a variety of benchmarks.
>
>
> We hope our responses answer your questions and that you can recommend acceptance.

---

> > ### Comment · Reviewer_j8uk · 2024-11-25
> >
> > I appreciate your efforts to improve the paper. However, I must maintain my original assessment for two reasons. First, the additional experiments, while valuable, are still limited to closely related computer vision tasks - given your method's simplicity and wide applicability, demonstrating its utility across more diverse domains would have been more convincing. Second, the concern about novelty remains unaddressed. The response attempts to distance itself from related approaches rather than constructively analyzing and positioning the work within the existing literature, which limits the scientific impact of this paper.

---

> > > ### Author Response · Authors · 2024-11-29
> > > **Response -- Novelty, Related Work and Experiments**
> > >
> > > We sincerely thank you for engaging with our rebuttal. While we appreciate your perspective, we respectfully disagree with the characterization of our contributions and the scope of the additional experiments.
> > >
> > > **On Task Diversity**:
> > > Our focus has been on classification tasks, where we demonstrated the broad applicability of our approach across multiple computer vision tasks, including image classification, video classification, multi-label classification, and semantic segmentation, as well as its extension to NLP domain. Each task requires a unique approach to estimating an aggregate sample-level confidence score.  Importantly, we demontrated how to construct approaches to do so for all these diverse tasks without making any assumptions about how models are trained or using multiple forward passes, which differentiates our approach.
> > >
> > > While we recognize the potential value of extending this work to more complex domains such as generative models, such an expansion is beyond the current scope and will form an important focus for our future work.
> > >
> > > **On Related Work and Novelty**:
> > > We sincerely apologize if our responses appeared as an attempt to distance ourselves from the suggested related works. That was not our intention. On the contrary, we acknowledge the relevance of these works and have expanded the Related Work section (see Lines 169–178 in the revised paper) to include quantization methods, following the suggestion of other reviewers.
> > >
> > > Furthermore, even in the original submission, we explicitly mentioned relevant parallels, such as speculative decoding, in the discussion section. For example, we wrote:
> > > “Little-Big is a subtractive multi-pass algorithm that relies on a good decomposition of problems and solves each part with the least compute, not unlike Speculative Decoding in language modeling (Leviathan et al., 2023).”
> > >
> > > Our objective in the rebuttal was to acknowledge these relevant works while also drawing clear distinctions, which is the purpose of any robust related work section. We see this as an essential part of positioning our contribution within the literature rather than as distancing from prior efforts.
> > >
> > > We remain open to addressing specific suggestions or concerns. Thank you once again.

---

### Official Review · Reviewer_rMyQ · 2024-11-01

**Soundness:** 2
**Presentation:** 3
**Contribution:** 1
**Rating:** 3
**Confidence:** 3

**Summary:**

This paper proposes a simple yet effective two-pass algorithm called "Little-Big" to speed up image classifiers.  The core idea is to leverage a smaller, less computationally expensive "Little" model to pre-screen input samples. Only samples for which the Little model exhibits low confidence are then passed to a larger, more accurate "Big" model.  The paper claims that this approach significantly reduces the computational cost (measured by Multiply-Accumulate operations or MACs) for a variety of model architectures and scales, without sacrificing accuracy on ImageNet-1K and other datasets. The authors demonstrate MACs reductions of up to 80% while maintaining or even improving accuracy compared to the single Big model baseline. They also argue that this approach is more effective than existing model compression and adaptive computation methods.

**Strengths:**

- The proposed Little-Big algorithm is conceptually straightforward and easy to implement. It requires minimal modifications to existing models and training pipelines.
- The paper demonstrates significant MACs reduction across a range of model architectures (CNNs, transformers, hybrids) and scales, suggesting broad applicability.
- Experiments are conducted on multiple datasets (ImageNet-1K, ImageNet-ReaL, ImageNet-V2) to evaluate the robustness and generalizability of the method.
- The Little-Big approach addresses a critical issue in deploying large vision models: their high computational cost. The proposed method offers a practical solution for model compression without retraining or complex modifications.

**Weaknesses:**

# Major
- The method seems to rely on finding an optimal threshold T on the test set (Imagenet validation set) to determine which samples are passed to the Big model. This raises concerns about potential overfitting to the validation set and its impact on generalization performance. Results should be provided using a threshold determined on the training or a held-out portion of the validation set to address this concern.
- The paper could benefit from a more comprehensive discussion of related work, particularly in areas like cascade models and dynamic inference methods.  Specifically, work on early-exit models [1] and confidence-based dynamic routing [2] appears closely related and should be discussed. This would help to better contextualize the novelty and contributions of the proposed approach. UPDATE: {See my response below.}
- Experiments solely focuses on ImageNet dataset. More experiments needed to understand the robustness of the proposed method.
- I also find it difficult to parse the results presented in huge tables: specifically Table 3: there are multiple baselines for DeiT models. Are you comparing results with different baseline accuracies?

[1] https://github.com/txsun1997/awesome-early-exiting?tab=readme-ov-file
[2] https://arxiv.org/pdf/2102.04906

# Minor
- [L132] I can't see the definition of "w and l".
- [Section 2.3] Quantization is a key method for compression and not mentioned here. Also Mixture of Depths (https://arxiv.org/abs/2404.02258)
- The paper's use of "hardness" and its relationship to model confidence is not always clear. In some sections, low confidence is equated with hardness, while in others, the opposite is implied. This needs clarification. For example [L199] "which allows us to approximate a "hardness" axis with prediction confidence." hardness means low confidence, no?

**Questions:**

- Please clarify the relationship between "hardness" and model confidence. Is low confidence always indicative of a hard sample, or are there cases where this assumption does not hold?
- Can you provide results where the threshold T is determined using only the training set or a held-out portion of the validation set? This would help to assess the potential for overfitting to the validation set and the generalizability of the method.

---

> ### Author Response · Authors · 2024-11-22
> **Part 1 of our response**
>
> **Dataset Diversity and Additional Experiments to Demonstrate Robustness**
>
> While the main paper focuses on the applicability of Little-Big to image and video classification (Kinetics-400), we have now extended our experiments, as suggested by the reviewer, to include additional datasets and settings to further demonstrate the robustness of our method.
>
> We kindly request the reviewer to check our summary response on our expanded experimental evaluation (namely, zero-shot image classification, text classification, multi-label classification, and a pixel-level semantic segmentation). Details of the newly added experiments can be found in Appendix A.1 of the revised paper.
>
> **Threshold Selection and Generalization Concerns**-
>
> In Section 4.2 of the main paper, we address this question in detail. Below is a summary of our findings:
>
>  * **Threshold Determination on Smaller Datasets**:
> We evaluate the generalizability of the threshold T by determining it using a smaller validation set (e.g., ImageNet-V2) and transferring it directly to the larger ImageNet-1K dataset. We opted for this approach to ensure comparability with baselines evaluated on the full ImageNet-1K validation split. As shown in Appendix Figure 7, even when T is determined on ImageNet-V2, the Little-Big framework achieves significant MACs (>80%) reductions with no performance drop on ImageNet-1K.
>
> * **Cross-Dataset Threshold Transfer**:
> We also examine the robustness of thresholds determined on the ImageNet-1K validation set by applying them directly to ImageNet-ReaL and ImageNet-V2. On both datasets, Little-Big achieves consistent MACs savings (>75%) with marginal accuracy losses of 0.04% and 0.07%, respectively.
>
> * **Threshold Determination for Non-ImageNet Tasks**:
> In our newly added experiments (Section 7 and summary response), thresholds were determined using only 20% of the held-out validation data while performance and compute savings were then evaluated on the remaining 80%. Our results clearly demonstrate the generalizability of our approach across diverse modalities and tasks.
>
>  **Related Work**
>
> We have expanded the related work section in the paper to include the additional methods suggested by the reviewer. Below, we summarize the key differences between these approaches and our proposed method:
> * **Early-Exit Models**: Early-exit models reduce computation by allowing samples to exit at intermediate stages of the model pipeline. However, implementing these models is non-trivial and highly architecture-dependent. They also require training with specific objectives tailored to the early-exit mechanism. In contrast, Little-Big is architecture-agnostic, completely post hoc, and broadly applicable across multiple tasks, model families, and domains (e.g., images and text). Furthermore, we compare with Dynamic-Vit and A-ViT which are early-exit like models in Section 4.4 and demonstrate the benefits of Little-Big over these approaches.
> * **Confidence-Based Dynamic Routing**: While similar to our approach in principle, confidence-based dynamic routing typically requires task-specific integration and training modifications. Little-Big differs itself by offering a simple thresholding mechanism that works across a wide range of pretrained models without requiring additional training.
> * **Quantization*: Quantization is indeed a relevant technique for model compression. While the focus of Little-Big is on inference-time compute savings without retraining, we acknowledge that quantization could further complement Little-Big. We have incorporated a short discussion of quantization in Section 2.3 and positioned it as an complementary method that can be integrated with Little-Big for additional gains. Furthermore,we highlight that the benefits of quantization are hardware dependent while Little-Big is a hardware independent approach.
>
> **Baselines**
> Thank you for this question.  In the A-ViT paper by Yin et al., the authors report the baseline DeiT performance as 78.9%, whereas the original DeiT paper reports a performance of 79.81% for the same model. Additionally, we confirmed the 79.81% performance through the publicly released DeiT-S checkpoint. Since it was not immediately clear which of these values to report, and to ensure fairness when comparing against other methods, we initially included both. However, we acknowledge that this choice could cause confusion for the reader. To address this, we have revised the paper to only include the 79.81% baseline from the original DeiT paper for clarity and consistency.
> We also made similar changes in Table-3 with  DeiT-B architecture to improve readability.

---

> > ### Author Response · Authors · 2024-11-22
> > **Part 2 of our response**
> >
> > **Definition of w and l**
> > We apologize for the confusion. While the definitions of \( w \) and \( l \) were provided in the caption of Table 1 in the main paper, we have now added a clarifying statement in the revised version after equation 2 to explicitly define \( w \) as the **width** and \( l \) as the **depth** of the model respectively.
> >
> >
> > **Hardness and confidence**
> >
> >
> > Thank you for this question and apologies for the confusion. In this paper, we correlate hardness with low confidence i.e,  the lower the confidence of the Little model on a sample, the harder that sample is. We have corrected Line 199 as follows: "This allows us to approximate a hardness axis, where harder samples correspond to predictions with lower confidence." We have also made this change in the revised paper.

---

> ### Comment · Reviewer_rMyQ · 2024-12-01
> **Response**
>
> Thank you the author for their response. I was about to raise my scores since my concerns were addressed except I wanted to check the literature one more time from the survey I've shared earlier. I would have expected authors to do the same especially after my review. Looks like they either didn't do it, or somehow missed this paper:
>
> https://dl.acm.org/doi/pdf/10.5555/2830840.2830854
>
> From what I see it is pretty much this work, except it uses networks from 10 years ago. Given the existence of this work and lack of comparison in the current paper I lower my score.

---

> > ### Author Response · Authors · 2024-12-01
> > **we are withdrawing our submission. We sincerely apologize for our oversight.**
> >
> > We thank you for bringing this paper to our attention. We sincerely apologize for missing it during our literature review and this was a complete oversight on our part.
> > In light of its relevance to our work, we will withdraw our submission to reassess our contributions and re-submit in the future.

---

### Official Review · Reviewer_kjCS · 2024-11-04

**Soundness:** 3
**Presentation:** 3
**Contribution:** 2
**Rating:** 5
**Confidence:** 3

**Summary:**

The paper proposes a method named Little-Big to accelerate image classification with neural networks. Little-Big uses a light-weight model to quickly classify all of the samples and selects the "hard" samples which get low confidence behind the threshold. Then, it uses a large model to update the prediction for each hard sample. Little-Big can significantly reduce the inference cost and latency for many advanced large classification models without sacrificing the accuracy. The authors provide many experiments with different pairs of large and small models to validate the effectiveness of Little-Big.

**Strengths:**

(1) The motivation and method of Little-Big is very simple and straightforward.

(2) It seems that Little-Big is very easy to implement. In addition, Little-Big is model-agnostic which can be applied to models with different scales and architectures.

(3) Little-Big can accelerate a pre-trained model without introducing additional training cost.

**Weaknesses:**

(1) Lack of novelty. As the authors say, Little-Big is an embarrassingly simple method, which adopts a large model and a light-weight model for image classification. It's the major advantage but also the major disadvantage of Little-Big. Many previous works share the similar motivation with Little-Big which uses different networks for accelerating, such as early existing and speculative decoding as you mentioned in the paper. While these works mostly include specific and delicate designs. I understand that Little-Big is very simple, but i don't think it's novel.

(2) The proposed method only focuses on the classification tasks. While the authors provide the example about how to extend it to video classification, it's hard to directly apply the method for other popular tasks (e.g., object detection and segmentation), which limits its use.

(3) The authors could include more classification tasks to further prove the generation ability of Little-Big, such as multi-label classification, binary classification.

**Questions:**

(1) The authors could also plot a figure for top-1 accuracy vs. latency.

(2) How long does it take to load and unload models compared with the inference latency for each batch? I'm wondering whether the method can be used for online inference.

---

> ### Author Response · Authors · 2024-11-22
> **Conducted experiments on all the settings suggested.**
>
> We thank you for your review.
>
> **Additional Experiments** We have now included additional experiments on all the tasks you suggested including - binary classification, multi-label classification, Zero-shot classification and semantic segmentation. Please refer to the our summary response and Sections A.1 in the updated paper.
>
> In summary, across all the tasks, Little-Big consistently achieves significant MAC savings while comparable comparbale performance compared to the Big model.
>
>
> **Accuracy vs Latency and Online Classification**
>
> Thank you for raising this important point.
>
> In the table below, we provide a comparison of change in the top-1 accuracy versus latency in a memory-constrained setup. In this scenario, we assume that both models cannot simultaneously fit in memory, requiring the Little model to be removed when the Big model is loaded. Even under these online classification constraints, Little-Big achieves high throughput and low latency while preserving the performance of the Big model.
>
> It is worth noting that the latency and throughput numbers would further improve if the hardware had sufficient memory to fit both models simultaneously.
>
> For more details, please refer to Section A.4 in the Appendix, titled "Benchmarking Throughput and Latency."
>
>
>
> | Big Model             | Little Model | Δ Acc   | Δ GMACs | Throughput (samples/s) | Δ Throughput | Latency (ms) | Δ Latency |
> |-----------------------|--------------|---------|---------|-------------------------|--------------|--------------|-----------|
> | EfficientNet-B7-600   | None         | --      | --      | 35.5                   | --           | 28.2         | --        |
> |                       | B0-224       | +0.01   | -25%    | 54.0                   | +52%         | 18.5         | -34%      |
> |                       | B1-240       | +0.01   | -13%    | 39.9                   | +12%         | 25.1         | -11%      |
> |                       | B2-288       | +0.00   | -59%    | 76.6                   | +116%        | 13.1         | -54%      |
> |                       | B3-300       | +0.02   | -61%    | 86.4                   | +144%        | 11.6         | -59%      |
> |                       | B4-380       | +0.01   | -81%    | 155.2                  | +338%        | 6.4          | -77%      |
> |                       | B5-456       | +0.01   | -65%    | 97.1                   | +174%        | 10.3         | -64%      |
> |                       | B6-528       | +0.02   | -47%    | 62.4                   | +76%         | 16.0         | -43%      |
>
>
>
>
>
>
>
> ## **Novelty**
>
> As the ML community continues to produce increasingly large models with massive parameter counts, efficiently deploying them has become a significant challenge. Current inference optimization strategies, such as quantization or distillation, often require training new, smaller models or performing operations that result in performance drops. Additionally, these techniques fail to fully leverage the extensive ecosystem of models that is readily available (e.g., the EfficientNet family or user-submitted models on platforms like Torch or HuggingFace Hub).
>
> In this context, we believe that our study on speeding up large models using smaller models is an important direction of work. Though the proposed protocol is simple to implement, our work holds significant practical value since
> - our approach completely **post-hoc**, requiring no re-training.
> - our approach is entirely **model- and architecture-agnostic**, allowing seamless integration with a variety of models.
> - ours is the first work to systematically study and benchmark its utility in improving inference efficiency across a variety of tasks and model architectures.
>
> By leveraging the growing diversity of pre-existing models across frameworks, Little-Big enables users to mix and match architectures (e.g., pairing models from different families such as EfficientNet and ViTs, BERTs and T5s). This flexibility ensures that one can adopt state-of-the-art models without the burden of additional training or specialized pipelines, and obtain significant compute savings and latency reductions without sacrificing accuracy. We argue that the framework's simplicity, adaptability, and compatibility with existing models make it a highly practical solution for real-world use cases.
>
>
> We sincerely hope the new experiments and our responses answer your questions and you can champion our paper.

---

### Author Response · Authors · 2024-11-22
**Summary Response - Additional experiments- Part 1**

We thank all reviewers for their insightful and constructive feedback.
### **Extensions and New Experiments**

A recurring suggestion was to extend the Little-Big framework and provide additional experimental validation of its effectiveness on other tasks. Following the suggestion, we have now expanded our empirical evaluation with four additional experiments (both vision and NLP tasks). While we present high-level summaries here, additional experimental details are provided in Appendix A.1 of the revised paper.

1. **Multi-Modal Zero-Shot Classification Tasks**:
   We study the use of our proposed approach in multi-modal zero-shot classification by accelerating the large CLIP model (ViT-L-14, 427M parameters) using a smaller CLIP model (ViT-B-32, 149M parameters). We considered four standard zero-shot evaluation benchmarks (Food-101, Flowers-102, Describable Textures, and SUN397). Our results corrobate the key finding in the original paper that our Little-Big protocol consistently achieves non-trivial reduction in MACs while producing peformance comparable to the big model. (on average, we notice a 40% reduction in MACs with only a 0.35% drop in accuracy).


| **Dataset**   | **Little Model Acc (%)** | **Big Model Acc (%)** | **Little-Big Acc (%)** | **$\Delta$ GMACs (%)** |
|---------------|---------------------------|------------------------|-------------------------|------------------------|
| SUN397        | 54.09                    | 58.50                 | 58.44                  | -49.91                |
| Food 101      | 78.42                    | 89.79                 | 89.41                  | -45.16                |
| DTD           | 31.86                    | 37.44                 | 37.44                  | -31.20                |
| Flowers 102   | 53.29                    | 66.35                 | 65.31                  | -33.49                |


2. **Multi-label Classification**:
   We also evaluated the Little-Big framework using a multi-label classification experiment with the popular CelebA facial attribute benchmark. Here, we used the same model configurations from the previous experiment -- "ViT-B-32" and "ViT-L-14" respectively. To estimate the model's confidence for a class, we measure the absolute difference of the prediction probability (after applying the sigmoid function) from 0.5. Note, 0.5 represents the highest uncertainity in a binary classification setting.Subsquently, we aggregate (i.e., average) the confidence scores across all classes to obtain a sample-level confidence estimate. Following the common experimental protocol adopted in this study, we determined the confidence threshold using a randomly chosen 20% of the held-out validation dataset and evaluated performance on the remaining 80% of the validation set.

   As shown in the table below, we achieve significant computational savings while incurring only a small drop in F1 score, thus evidencing the utility of the Little-Big protocol even in multi-label classification.

| Little Model F1 (%) | Big Model F1 (%) | Little-Big F1 (%) | MACs Reduction (%) |
|:--------------------:|:----------------:|:------------------:|:-------------------:|
|         60.44       |       64.31      |        63.28       |        40.10        |

---

> ### Author Response · Authors · 2024-11-22
> **Summary Response - Additional experiments- Part 2**
>
> 3. **Semantic Segmentation**
>
>   Next, we applied Little-Big to a more challenging pixel-level task -- semantic segmentation. For this experiment, we used the PyTorch—DeepLabV3 (MobileNet backbone, 11M parameters) as the Little model and the FCN (ResNet-101 backbone, 54M parameters)  as the Big model. The evaluation was conducted using a subset of COCO with 20 categories overlapping with the Pascal VOC dataset. In this setup, we esimate the sample-level confidence as the lowest confidence among all superpixels ( superpixels are obtained using the SLIC algorithm), wherein the superpixel confidence was measured as the average of max. softmax probabilities from all pixels constituting the superpixel.
>
>   The result below further demonstrates the benefits of our approach.
>
>
> | Little Model mIOU (%) | Big Model mIOU (%) | Little-Big mIOU (%) | MACs Reduction (%) |
> |:--------------------:|:----------------:|:------------------:|:-------------------:|
> |         60.3       |       63.7      |        63.0       |        39.2       |
>
> 4. **NLP tasks**
>
>   Finally, as requested by reviewers, we extended the Little-Big framework to NLP tasks, specifically focusing on text classification. We conducted experiments on the IMDB sentiment (binary) classification benchmark using DistilBERT (66M parameters) as the Little model and GPT-2 (124 parameters) as the Big model.  The results, presented below, show that our approach achieves over 58% MACs reduction while maintaining the performance of the Big model. This demonstrates that Little-Big is not limited to vision tasks but is also effective in natural language processing domains.
>
> | Little Model Acc (%) | Big Model Acc (%) | Little-Big Acc (%) | MACs Reduction (%) |
> |:--------------------:|:----------------:|:------------------:|:-------------------:|
> |         92.8       |       93.5     |        93.51      |        58.75       |

---

> > ### Author Response · Authors · 2024-11-29
> > **Summary response -- Comprehensive Evaluation of Little-Big Model Choices**
> >
> > **Comprehensive Evaluation and Interactive Results Through Streamlit**
> >
> > In response to feedback requesting a broader evaluation, we have expanded our experiments to include **all possible little-big model pairs from a pool of 63 models, resulting in 1,953 combinations**. This thorough evaluation ensures that our method is rigorously tested across a wide variety of settings.
> >
> > To enhance accessibility and demonstrate the scalability of our approach, we have created an interactive tool available at https://littlebigpaper.streamlit.app/. This platform allows users to select any big and little model from dropdown menus and view the detailed results for their chosen combination, including GMAC savings, accuracy comparisons, and more.

---

> > > ### Author Response · Authors · 2024-12-01
> > > **We are withdrawing our submission.**
> > >
> > > Hello all,
> > >
> > > We sincerely thank all reviewers for their thoughtful feedback and for engaging with our work. During the review process, a reviewer brought to our attention a prior paper that bears significant relevance to our submission, which we regretfully overlooked during our literature review. This was a major oversight on our part.
> > >
> > > In light of this, we have decided to withdraw our submission to reassess our contributions and re-submit in the future.
> > >
> > > We deeply appreciate the time and effort each reviewer has invested in critiquing our work and we once again thank you for your valuable feedback.

---

### Note · Authors · 2024-12-01

**Comment:**

We sincerely thank all reviewers for their thoughtful feedback and for engaging with our work. During the review process, a reviewer brought to our attention a prior paper that bears significant relevance to our submission, which we regretfully overlooked during our literature review. This was a major oversight on our part.

In light of this, we have decided to withdraw our submission to reassess our contributions and re-submit in the future.

We deeply appreciate the time and effort each reviewer has invested in critiquing our work and we once again thank you for your valuable feedback.

**Withdrawal Confirmation:**

I have read and agree with the venue's withdrawal policy on behalf of myself and my co-authors.